# Test-time Correlation Alignment

**Linjing You** [*1] **Jiabao Lu** [*1] **Xiayuan Huang**[† 2]

## Abstract

Deep neural networks often degrade under distribution shifts. Although domain adaptation offers a solution, privacy constraints often prevent access to source data, making Test-Time Adaptation (TTA)—which adapts using only unlabeled test data—increasingly attractive. However, current TTA methods still face practical challenges: (1) a primary focus on instance-wise alignment, overlooking CORrelation ALignment (CORAL) due to missing source correlations; (2) complex backpropagation operations for model updating, resulting in overhead computation and (3) domain forgetting. To address these challenges, we provide a theoretical analysis to investigate the feasibility of **T**est-time **C**orrelation **A**lignment (**TCA**), demonstrating that correlation alignment between high-certainty instances and test instances can enhance test performances with a theoretical guarantee. Based on this, we propose two simple yet effective algorithms: LinearTCA and LinearTCA$^+$. LinearTCA applies a simple linear transformation to achieve both instance and correlation alignment without additional model updates, while LinearTCA$^+$ serves as a plug-and-play module that can easily boost existing TTA methods. Extensive experiments validate our theoretical insights and show that TCA methods significantly outperforms baselines across various tasks, benchmarks and backbones. Notably, LinearTCA achieves higher accuracy with only 4% GPU memory and 0.6% computation time compared to the best TTA baseline. It also outperforms existing methods on CLIP over 1.86%. Code: https://github.com/youlj109/TCA.

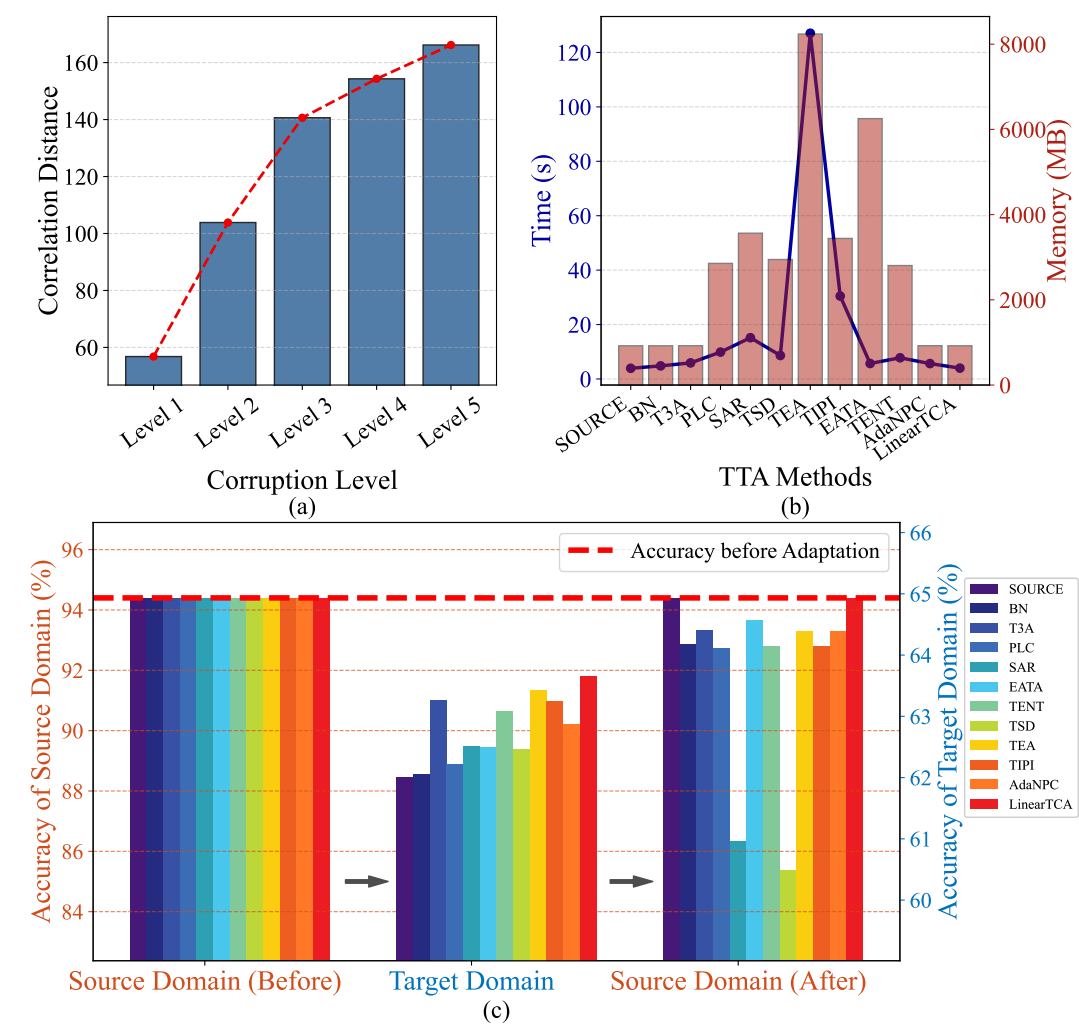

*Figure 1.* Illustration of key limitations in existing TTA methods. (a) Correlation distance increases with domain shifts. (b) Computation time and peak GPU memory usage on CIFAR-10-C, showing high overhead of existing methods. (c) Source domain performance after test-time adaptation, revealing challenges in retaining source knowledge.

## 1. Introduction

Deep neural networks (DNNs) have significantly advanced numerous tasks in recent years (LeCun et al., 2015; Jumper et al., 2021; Silver et al., 2016) when the training and test data are independent and identically distributed (i.i.d.). However, the i.i.d. condition rarely holds in practice as the data distributions are likely to change over time and space (Fang et al., 2020; Wang & Deng, 2018). This phenomenon, known as the out-of-distribution (OOD) problem or distribution shift, has been extensively investigated within the context of domain adaptation (DA) (You et al., 2019; Zhou et al., 2022; Liang et al., 2024). Among various DA methods, CORrelation ALignment (CORAL) (Sun et al., 2017; Sun & Saenko, 2016; Cheng et al., 2021a) has been proven to be an effective and "frustratingly simple" paradigm, which aligns the feature distributions of the source and target domains at a feature correlation level rather than merely aligning individual instances.

However, DA methods are practically difficult when pre-

[*]Equal contribution [1]Institute of Automation, Chinese Academy of Sciences [2]College of Science, Beijing Forestry University. Correspondence to: Xiayuan Huang <huangxiayuan@bjfu.edu.cn>, Linjing You <youlinjing2023@ia.ac.cn>.

*Proceedings of the 42nd International Conference on Machine Learning*, Vancouver, Canada. PMLR 267, 2025. 

trained models are publicly available but the training data and training process remain inaccessible due to privacy and resource restrictions (Liang et al., 2024). To address such a source-inaccessible domain shifts task at test time, test-time adaptation (TTA) (Gong et al., 2024; Su et al., 2024a;b; You et al., 2025) has emerged as a rapidly progressing research topic. Although some recent attempts have been made to handle this task, current TTA methods still face several limitations:

Firstly, overlooking feature correlations: Most existing TTA methods focus on instance-wise alignment (Wang et al., 2023; Nguyen et al., 2023; Wang et al., 2020) that only capture central of the instances while neglecting the correlations between features. For example, relationships between edge and texture features can vary significantly across domains. Let's consider a simple test on the CIFAR-10-C dataset (Hendrycks & Dietterich, 2019) to show the relationship between feature correlation and domain shift. As shown in Figure 1a , the correlation distance (see Section 2.2) of ResNet-18 (He et al., 2016) embedding are computed with an increasing corruption level from 1 to 5. It illustrates that as domain shifts increase, the changes in feature correlation also increase.

Secondly, overhead computation: Current TTA methods often rely on computationally expensive backpropagation for each test sample to update models (Sun et al., 2020; Wang et al., 2020; Goyal et al., 2022; Bartler et al., 2022). However, many applications are deployed on edge devices, such as smartphones and embedded systems (Niu et al., 2024), which typically lack the computational power and memory capacity required for such intensive calculations. As a result, backpropagation-based TTA methods are limited in their applicability on these edge devices. In Figure 1b, we illustrate the computation time and maximum GPU memory usage of different TTA methods on the CIFAR-10-C dataset. Compared to the non-adaptive source model (ERM(Vapnik, 1999)), most TTA methods show a dramatic increase in both items.

Lastly, domain forgetting: Another drawback of backpropagation-based TTA methods is that they often lead to model updating, which gradually loses the prediction ability of the source or training domain (Niu et al., 2024; Zhang et al., 2023). As illustrated in Figure 1c, after adaptation on test domain, the performance of most methods declines when return to the source domain, indicating that existing TTA approaches struggle to retain knowledge of the source domain.

To address the above challenges, applying the "effective and frustratingly simple" CORAL method to TTA appears intuitive—but the lack of source data makes it highly challenging. We thus explore the feasibility of **T**est-time **C**orrelation **A**lignment (**TCA**) by posing key questions: *(1)Can we construct a pseudo-source correlation that approximates the true source correlation? (2) Can this enable effective TTA?* We provide a theoretical analysis showing that aligning correlations between high-certainty and test instances improves test-time performance with guarantees. Based on this, we propose two simple yet effective methods: LinearTCA and LinearTCA⁺. Specifically, we first compute the "pseudo-source correlation" by using $k$ high-certainty instances. Then, LinearTCA aligns correlation through simple linear transformations of embeddings without model updates, resulting in minimal computation and keeping source domain knowledge. While LinearTCA⁺ serves as a plug-and-play module that can easily boost existing TTA methods.

**Main Findings and Contributions**: (1) We introduce a novel and practical paradigm for TTA, termed Test-time Correlation Alignment (TCA). The construction of the pseudo-source correlation and the adaptation effectiveness are theoretically guaranteed. (2) Based on our analysis, we develop two simple yet effective methods—LinearTCA and LinearTCA⁺—to validate TCA's effectiveness and its plug-and-play potential with other TTA approaches. (3)We conduct comprehensive experiments to validate our theoretical insights and compare performance across diverse benchmarks, backbones, and tasks, evaluating accuracy, efficiency, and resistance to forgetting. Results show that LinearTCA achieves outstanding performance, while LinearTCA⁺ robustly boosts other TTA methods under various conditions. (4) Further in-depth experimental analysis reveals the effective range of LinearTCA and provides valuable insights for future work.

## 2. Preliminary and Problem Statement

We briefly revisit TTA and CORAL in this section for the convenience of further analyses, and put detailed related work discussions into Appendix A due to page limits.

### 2.1. Test Time Adaptation (TTA)

In the test-time adaptation (TTA) (Tan et al., 2024; Yuan et al., 2023) scenario, it has access only to unlabeled data from the test domain and a pre-trained model from the source domain. Specifically, let $D_s = \{(x_s^i, y_s^i)\}_{i=1}^{n_s} \sim \mathbb{D}_s$ represent the labeled source domain dataset, where $(x_s^i, y_s^i)$ is sampled i.i.d from the distribution $\mathbb{D}_s$ and $n_s$ is the number of the total source instances. The model, trained on the source domain dataset and parameterized by $\theta$, is denoted as $h_\theta(\cdot) = g(f(\cdot)) : \mathcal{X}_s \rightarrow \mathcal{Y}_s$, where $f(\cdot)$ is the backbone encoder and $g(\cdot)$ denotes the decoder head. During testing, $h_\theta(\cdot)$ will perform well on in-distribution (ID) test instances drawn from $\mathbb{D}_s$. However, given a set of out-of-distribution (OOD) test instances $D_t = \{x_t^i\}_{i=1}^{n_t} \sim \mathbb{D}_t$ and $\mathbb{D}_t \neq \mathbb{D}_s$, the prediction performance of $h_\theta(\cdot)$ would

decrease significantly. To this end, the goal of TTA is to adapt this model $h_\theta(\cdot)$ to $D_t$ without access to $D_s$. For each instance $x_t^i \in \mathcal{X}_t$, let the output of encoder $f(\cdot)$ and decoder $g(\cdot)$ be denoted as $z_t^i = f(x_t^i) \in \mathbb{R}^d$ and $p_t^i = g(z_t^i) \in \mathbb{R}^c$, respectively, where $d$ is the dimension of the embeddings and $c$ is the number of classes in a classification task. When encountering an OOD test instance $x_t^i$, existing TTA methods (Wu et al., 2024; Sinha et al., 2023; Lee et al., 2024; Yuan et al., 2023) typically minimize an unsupervised or self-supervised loss function to align the embedding $z_t^i$ or prediction $p_t^i$, thereby updating the model parameters $\theta$:

$$\min_{\tilde{\theta}} \mathcal{L}(z_t^i, p_t^i, \theta), \quad x_t^i \sim \mathbb{D}_t \tag{1}$$

where $\tilde{\theta} \subseteq \theta$ is a proper subset of $\theta$ involved in the update, such as the parameters of the batch normalization (BN) layers (Schneider et al., 2020; Su et al., 2024c) or all parameters. Generally, the TTA loss function $\mathcal{L}(\cdot)$ can be formulated by nearest neighbor information (Zhang et al., 2023; Hardt & Sun, 2023; Jang et al., 2022), contrastive learning (Wang et al., 2023; Chen et al., 2022), entropy minimization (Wang et al., 2020; Niu et al., 2022), etc.

### 2.2. Correlation Alignment (CORAL)

The aim of correlation alignment (CORAL) (Sun et al., 2017; Cheng et al., 2021a; Sun & Saenko, 2016; Sun et al., 2016; Das et al., 2021; Rahman et al., 2020b) is to minimize the distance of the second-order statistics (covariance) between the source and test features. Specifically, let $Z_s = \{z_s^i\}_{i=1}^{n_s} \in \mathbb{R}^{n_s \times d}$ denotes the feature matrix from the source domain, and $Z_t = \{z_t^i\}_{i=1}^{n_t} \in \mathbb{R}^{n_t \times d}$ denotes the feature matrix from the test domain. CORAL computes the covariance matrices of the source features $Z_s$ and test features $Z_t$, and aligns correlation by minimizing the Frobenius norm of their two covariance matrices. The covariance matrix is computed as below:

$$\Sigma = \frac{1}{n-1}(Z^T Z - \frac{1}{n}\mathbf{1}_n Z^T Z \mathbf{1}_n) \tag{2}$$

the correlation distance is then given by (Sun & Saenko, 2016):

$$\mathsf{d}(\Sigma_s, \Sigma_t) = \frac{1}{4d^2}\|\Sigma_s - \Sigma_t\|_F^2 \tag{3}$$

where $\Sigma_s$ and $\Sigma_t$ are the covariance matrices of the source and test domains, respectively, and $\mathbf{1}$ is a column vector with all elements equal to 1 to perform mean-subtraction. $\|\cdot\|_F$ represents the Frobenius norm.

### 2.3. Problem Statement

Existing TTA methods suffer from overlooking feature correlation, overhead computation and domain forgetting. Research and practice have demonstrated that CORAL is both effective and “frustratingly easy” to implement on DA. Since TTA is a subfield of DA, it is a natural extension to apply CORAL within TTA frameworks. However, due to privacy and resource constraints in TTA, it is impossible to compute the source correlation. This limitation hinders the application of CORAL in such real-world scenarios, i.e. test-time correlation alignment (TCA).

## 3. Theoretical Studies

In this section, we conduct an in-depth theoretical analysis of TCA based on domain adaptation and learning theory. We focus on two key questions: *(1) Can we construct a “pseudo-source correlation” to approximate the original source correlation? (2) Can TCA based on this pseudo-source correlation enable effective TTA?* Before discussing the main results, we first state some necessary assumptions and concepts. Missing proofs and detailed explanations are provided in Appendix B.

**Definition 3.1.** ***(Classification error and empirical error)*** Let $\mathcal{H}$ be a hypothesis class of VC-dimension $d_v$. The error that an estimated hypothesis $h_\theta \in \mathcal{H}$ disagrees with the groundtruth labeling function $l : \mathcal{X}_t \to \mathcal{Y}_t$ according to distribution $\mathbb{D}_t$ is defined as:

$$\epsilon(h_\theta, l) = \mathbb{E}_{x \sim \mathbb{D}_t}[|h_\theta(x) - l(x)|] \tag{4}$$

which we also refer to as the error or risk $\epsilon(h_\theta)$. The empirical error of $h_\theta \in \mathcal{H}$ with respect to a labeled dataset $D_s = \{(x_s^i, y_s^i)\}_{i=1}^{n_s} \sim \mathbb{D}_s$ is defined as:

$$\hat{\epsilon}(h_\theta) = \frac{1}{n_s}\sum_{i=1}^{n_s} |h_\theta(x_s^i) - y_s^i| \tag{5}$$

**Assumption 3.2.** ***(Strong density condition)*** Given the parameters $\mu^-, \mu^+, c_t, c_t^*, r_t > 0$, we assume that the distribution $\mathbb{D}_s$ and $\mathbb{D}_t$ are absolutely continuous with respect to the Lebesgue measure $\lambda[\cdot]$ in Euclidean space. Let $\mathcal{B}(x, r) = \{x_0 : \|x_0 - x\| \le r\}$ denote the closed ball centered at point $x$ with radius $r$. We further assume that $\forall\ x_t \sim \mathbb{D}_t$ and $r \in (0, r_t]$, the following conditions hold:

$$\lambda[\mathbb{D}_s \cap \mathcal{B}(x_t, r)] \ge c_t \lambda[\mathcal{B}(x_t, r)] \tag{6}$$

$$\lambda[\mathbb{D}_t \cap \mathcal{B}(x_t, r)] \ge c_t^* \lambda[\mathcal{B}(x_t, r)] \tag{7}$$

$$\mu^- < \frac{\partial \mathbb{D}_s}{\partial \lambda} < \mu^+; \quad \mu^- < \frac{\partial \mathbb{D}_t}{\partial \lambda} < \mu^+ \tag{8}$$

The strong density condition is commonly used when analyzing KNN classifiers (Audibert & Tsybakov, 2007; Cai & Wei, 2021). Recently, it has also been applied in the

test-time adaptation (Zhang et al., 2023). Intuitively, Assumption 3.2 requires that the divergence between $\mathbb{D}_s$ and $\mathbb{D}_t$ is bounded. When $c_t = 1$, for each $x_t \sim \mathbb{D}_t$, the neighborhood ball $\mathcal{B}(x_t, r)$ is completely contained within $\mathbb{D}_s$. In contrast, when $c_t = 0$, $\mathcal{B}(x_t, r)$ and $\mathbb{D}_s$ are nearly disjoint.

**Assumption 3.3.** ***(L-Lipschitz Continuity)*** Let $h_\theta(\cdot) = g(f(\cdot))$ be a estimated hypothesis on $\mathcal{H}$. We assume that there exists a constant $L$ such that $\forall\ x_1, x_2 \in D_s \cup D_t$, the encoder $f(\cdot)$ satisfies the following condition:

$$\|f(x_1) - f(x_2)\| \leq L\|x_1 - x_2\| \tag{9}$$

The assumption of L-Lipschitz continuity is frequently employed in the analysis of a model's adaptation capabilities (Mansour et al., 2009). It implies that the change rate of $f(\cdot)$ does not exhibit extreme fluctuations and is bounded by the constant $L$ at any point.

**Assumption 3.4.** ***(Taylor Approximation)*** Let $h_\theta(\cdot) = g(f(\cdot))$ be a $L$-Lipschitz Continuous hypothesis on $\mathcal{H}$. $z = f(x)$ and $p = g(z)$. We assume that there exists a constant $r^*$ such that $\forall\ x_1, x_2 \in D_s \cup D_t$, if $\|z_1 - z_2\| \leq r^*$, $p_2 = g(z_2)$ can be approximated using the first-order Taylor expansion at $z_1$ as follows:

$$p_2 = p_1 + J_g(z_1)(z_2 - z_1) + o(\|z_1 - z_2\|) \tag{10}$$

where $p_1 = g(z_1)$, $J_g(z_1)$ is the Jacobian matrix of $g$ evaluated at $z_1$, and $o(\|z_1 - z_2\|)$ represents the higher-order terms in the expansion.

It indicates that when the outputs $z_1$ and $z_2$ are close (i.e., their distance is within the radius $r^*$), the decoder can be well-approximated by a linear function at $z_1$.

### 3.1. Correlation of high-certainty test instances approximates the source correlation

We characterize the divergence of correlation between the pseudo-source and the source correlation in the following Theorem 3.5.

**Theorem 3.5.** *Let $h_\theta(\cdot) = g(f(\cdot))$ be an L-Lipschitz continuous hypothesis on $\mathcal{H}$. $\Omega := \bigcup_{x \in \mathbb{D}_t} \mathcal{B}(x, r^*)$ is the set of balls near the test data. We sample $k$ source instances from $\mathbb{D}_s \cap \Omega$ and $k$ test instances from $\mathbb{D}_t$ to obtain $[X_s, Z_s, P_s]$ and $[X_t, Z_t, P_t]$ by $h_\theta(\cdot)$, respectively. Per Assumption 3.2, Assumption 3.3 and Assumption 3.4, with a probability of at least $1 - \exp\left(-\frac{(c_t\mu^- \pi_{d_I} r^{d_I} n_s - 1)^2}{2c_t\mu^- \pi_{d_I} r^{d_I} n_s} + \log k\right)$, we have*

$$\|Z_t - Z_s\| \leq \frac{\|P_t - P_s\| + \|o(kr^*)\|}{\|J_g(Z_s)\|} \tag{11}$$

where $\pi_{d_I} = \lambda(\mathcal{B}(0,1))$ is the volume of the $d_I$ dimension unit ball and $d_I$ is the dimension of input $x$. Furthermore, considering the true source correlation $\Sigma_s = \mathbb{E}[\tilde{Z}_s^{\ T} \tilde{Z}_s]$ and the pseudo-source correlation $\hat{\Sigma}_s = \tilde{Z}_t^{\ T} \tilde{Z}_t$, where $\tilde{Z}_s$ and $\tilde{Z}_t$ are centered. With a probability of at least $\min(1 - \exp\left(-\frac{(c_t\mu^- \pi_{d_I} r^{d_I} n_s - 1)^2}{2c_t\mu^- \pi_{d_I} r^{d_I} n_s} + \log k\right), 1 - \delta)$, the correlation distance $\|\Sigma_s - \hat{\Sigma}_s\|$ is bounded by:

$$\|\Sigma_s - \hat{\Sigma}_s\|_F \leq$$
$$2\|Z_s\|_F\left(\frac{\|\hat{Y}_t - P_t\|_F + A}{\|J_g(Z_s)\|_F}\right) + \left(\frac{\|\hat{Y}_t - P_t\|_F + A}{\|J_g(Z_s)\|_F}\right)^2 + B \tag{12}$$

where $\hat{Y}_t$ is the one-hot encoding of $P_t$, $A = \|o(kr^*)\| + k\epsilon(h_\theta(X_t)) + k\epsilon(h_\theta(X_s))$ represents the output error of the sampled instances, and $B = \sqrt{\frac{\log(2/\delta)}{2k}}$ is the sampling error.

Theorem 3.5 implies the followings: (1) In Eq. (12), the terms $X_s$, $Z_s$, and $J_g(Z_s)$ remain unchanged with the same source data. The primary factor influencing the correlation distance $\|\Sigma_s - \hat{\Sigma}_s\|$ is prediction uncertainty $\|\hat{Y}_t - P_t\|_F$ and output error of the sampled instances $\epsilon(h_\theta(X_t))$. (2) Intuitively, previous studies (Gui et al., 2024; Niu et al., 2022; Yuan et al., 2024) empirically suggest that instances with higher output certainty have less output error. In other words, with a smaller divergence between the prediction $P_t$ and its one-hot encoding $\hat{Y}_t$, both uncertainty $\|\hat{Y}_t - P_t\|_F$ and error $\epsilon(h_\theta(X_t))$ will decrease, resulting in a smaller correlation distance. (3) Therefore, **a reasonable pseudo-source construction method is to select the $k$ test instances with the smallest $\|\hat{Y}_t - P_t\|_F$ values (i.e. high-certainty test instances) and compute their correlation matrix as pseudo-source correlation.**

### 3.2. Test-time correlation alignment reduces test classification error

In this section, we establish the TTA error bounds of hypothesis $h_\theta$ when minimizing the empirical error in the source data (Theorem 3.6) and examine the influence of using the pseudo-source correlation (Corollary 3.7), which further indicates factors that affect the performance of $h_\theta$.

**Theorem 3.6.** *Let $\mathcal{H}$ be a hypothesis class of VC-dimension $d_v$. If $\hat{h} \in \mathcal{H}$ minimizes the empirical error $\hat{\epsilon}_s(h)$ on $D_s$, and $h_t^* = \arg\min_{h \in \mathcal{H}} \epsilon_t(h)$ is the optimal hypothesis on $D_t$, with the assumption that all hypotheses are L-Lipschitz continuous, then $\forall \delta \in (0, 1)$, with probability with at least $1 - \delta$ the following inequality holds:*

$$\epsilon_t(\hat{h}) \leq \epsilon_t(h_t^*) + \mathcal{O}(\sqrt{\|\mu_s - \mu_t\|_F^2 + \|\Sigma_s - \Sigma_t\|_F^2}) + C$$

where $C = 2\sqrt{\frac{d_v log(2n_s) - \log(\delta)}{2n_s}} + 2\gamma$ and $\gamma = \min_{h \in \mathcal{H}}\{\epsilon_s(h(t)) + \epsilon_t(h(t))\}$. $\mu_s$, $\mu_t$, $\Sigma_s$ and $\Sigma_t$ denote the means and correlations of the source and test embeddings, respectively. We use $\mathcal{O}(\cdot)$ to hide the constant dependence.

For fixed $D_s$ and $D_t$, $\epsilon_t(h_t^*)$ and $C$ are constants, indicating that the primary factors affecting the performance of $h_\theta$ on the test data $D_t$ (i.e., $\epsilon_t(\hat{h})$) are $\|\mu_s - \mu_t\|_F^2$ and $\|\Sigma_s - \Sigma_t\|_F^2$. By aligning correlations during testing, which means reducing $\|\Sigma_s - \Sigma_t\|_F^2$, we can effectively decrease the model's classification error on the test data. Combining Theorem 3.5 with Theorem 3.6, the following corollary provides a direct theoretical guarantee that TCA based on pseudo-source correlation can reduce the error bounds on test data.

**Corollary 3.7.** *Let $\Sigma_s$, $\hat{\Sigma}_s$ and $\Sigma_t$ denote the source, pseudo-source and test correlation, respectively. Theorem 3.5 establishes the error bound between $\hat{\Sigma}_s$ and $\Sigma_s$, while Theorem 3.6 demonstrates that reducing the difference between $\Sigma_t$ and $\Sigma_s$ can decrease classification error on the test data. By applying the triangle inequality, we have:*

$$\|\Sigma_t - \Sigma_s\|_F = \|\Sigma_t - \hat{\Sigma}_s + \hat{\Sigma}_s - \Sigma_s\|_F \leq \|\Sigma_t - \hat{\Sigma}_s\|_F + \|\hat{\Sigma}_s - \Sigma_s\|_F \quad (13)$$

*Therefore, Theorem 3.6 can be rewritten as:*

$$\begin{aligned} &\epsilon_t(\hat{h}) \leq \\ &\epsilon_t(h_t^*) + \mathcal{O}(\sqrt{\|\mu_s - \mu_t\|_F^2 + \|\Sigma_s - \Sigma_t\|_F^2}) + C \leq \\ &\epsilon_t(h_t^*) + \mathcal{O}((\|\mu_s - \mu_t\|_F^2 + (2\|Z_s\|_F(\frac{\|\hat{Y}_t - P_t\|_F + A}{\|J_g(Z_s)\|_F}) \\ &+ (\frac{\|\hat{Y}_t - P_t\|_F + A}{\|J_g(Z_s)\|_F})^2 + B + \|\Sigma_t - \hat{\Sigma}_s\|_F)^2)^{1/2}) + C \end{aligned} \quad (14)$$

Corollary 3.7 indicates the followings: (1) Reducing the correlation distance between the test data and the pseudo-source, i.e., $\|\Sigma_t - \hat{\Sigma}_s\|_F^2$, can reduce the test classification error. The pseudo-source correlation $\hat{\Sigma}_s$ is computed by selecting $k$ instances from the test data with minimal uncertainty, measured by $\|\hat{Y}_t - P_t\|_F^2$. (2) Updating model parameters to decrease $\|\hat{Y}_t - P_t\|_F^2$ can further reduce the test error. (3) Additionally, minimizing the instance-wise distance $\|\mu_s - \mu_t\|_2^2$ can also contribute to reducing the test error, which is consistent with previous studies (Niu et al., 2022; Wang et al., 2023; 2020).

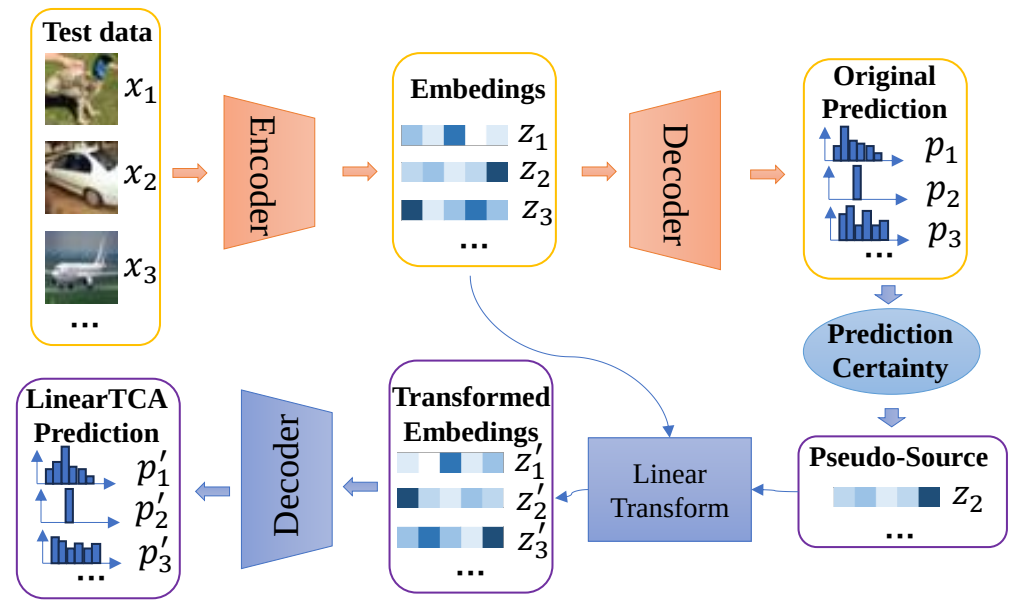

*Figure 2.* The pipeline of our proposed LinearTCA method. During testing, we first obtain original embeddings and predictions using the source model. Based on the certainty of the original predictions, we select a subset embeddings to form a "pseudo-source domain". A linear transformation is then applied to align the correlations of the original embeddings with those of the pseudo-source domain, ultimately producing the final predictions of LinearTCA. Notably, this process does not require updating any parameters of the original model.

**Remark.** Section 3.1 answers the first question that the feature correlation of high-certainty test instances from the pre-trained model can approximate the feature correlation of the source domain. Section 3.2 provides a theoretical guarantee that conducting correlation alignment between pseudo-source correlation and test correlation during TTA can effectively reduce the test error bound. These theoretical findings are further validated in Section 5.2.

## 4. The Test-time Correlation Alignment Algorithms

As illustrated in Figure 2, building on our theoretical findings, we propose two simple yet effective TCA methods: LinearTCA and LinearTCA$^+$. We start with detailing the construction of the pseudo-source correlation, followed by the implementation of LinearTCA and LinearTCA$^+$.

### 4.1. Pseudo-Source

For each instance $x_t^i$ arrives in test time, we first get embedding $z_t^i = f(x_t^i)$ and prediction $p_t^i = g(z_t^i)$. Per Theorem 3.5, we compute its prediction uncertainty $\omega_t^i = \|\hat{y}_t^i - p_t^i\|_F^2$, where $\hat{y}_t^i = onehot(argmax(p_t^i))$. We then temporarily store the pair $(z_t^i, \omega_t^i)$ in the Pseudo-Source bank $\mathcal{M} = \mathcal{M} \cup (z_t^i, \omega_t^i)$. Subsequently, $\mathcal{M}$ is updated based on its element count and confidence. The update rule is as follows:

$$\mathcal{M} = \begin{cases} \mathcal{M}, & \text{if } |\mathcal{M}| \leq k \\ \{(z_t^i, \omega_t^i) \mid \omega_t^i \leq \omega_{min}^k\}, & \text{else} \end{cases} \quad (15)$$

where $\omega_{min}^k$ represents $k$-th lowest uncertainty value in $\mathcal{M}$. Finally, the Pseudo-Source correlation can be calculated as

follows:

$$\hat{\Sigma}_s = \frac{1}{\hat{n}_s - 1}\left(\hat{Z}_s^T\hat{Z}_s - \frac{1}{\hat{n}_s}\mathbf{1}_{\hat{n}_s}\hat{Z}_s^T\hat{Z}_s\mathbf{1}_{\hat{n}_s}\right) \quad (16)$$

where $\hat{Z}_s = \{z_t^i | z_t^i \in \mathcal{M}\}$ and $\hat{n}_s = |\mathcal{M}|$.

### 4.2. Methods

**LinearTCA:** During testing, given the embeddings $Z_t$ and $\hat{Z}_s$ sampled from the test and pseudo-source domains, respectively, our objective is to minimize their correlation distance:

$$\mathcal{L}_{\text{LinearTCA}} = \left\|\Sigma_t - \hat{\Sigma}_s\right\|_F^2 \quad (17)$$

To achieve this alignment, we aim to obtain a suitable linear transformation $W$ as follows:

$$\min_W \left\|W^T\Sigma_t W - \hat{\Sigma}_s\right\|_F^2 \quad (18)$$

Setting $W^T\Sigma_t W = \hat{\Sigma}_s$ and applying eigenvalue decomposition, the closed-form solution for $W$ can be derived as [1]:

$$W = U_t\Lambda_t^{1/2}\hat{U}_s^T\hat{\Lambda}_s^{-1/2} \quad (19)$$

where $\hat{U}_s$ and $U_t$ represent the eigenvector matrices, $\hat{\Lambda}_s$ and $\Lambda_t$ are the corresponding diagonal eigenvalue matrices, respectively. The transformed embeddings of the test domain can then be computed as:

$$Z_t^{'} = (Z_t - \mu_t)\,W + \hat{\mu}_s \quad (20)$$

where $\mu_t$ and $\hat{\mu}_s$ denote the mean embeddings of $Z_t$ and $\hat{Z}_s$, respectively. As shown in Eq. (20), we also align the instance-wise shift $|\mu_s - \mu_t|$ by using $\hat{\mu}_s$. Finally, the predictions for the test domain after adaptation through LinearTCA are:

$$P_t^{'} = g(Z_t^{'}) \quad (21)$$

**LinearTCA⁺:** Since LinearTCA does not require parameter updates to the model, it can serve as a plug-and-play boosting module for TTA methods. Specifically, during a TTA method optimizes the original model $h_\theta$ to $h_{\tilde{\theta}}$ via Eq. (1), we can obtain the resulting embeddings $Z_{TTA}$ and predictions $P_{TTA}$. By applying the LinearTCA on $Z_{TTA}$ and $P_{TTA}$ with the same process from Eq. (15) to (21), the predictions of LinearTCA⁺ are obtained. More details on these methods are provided in Appendix C.

[1]To enhance the robustness of the results, we recommend using torch's automatic gradient descent method to mitigate potential instabilities associated with eigenvalue decomposition. For the following experiments, we implement this method with a fixed learning rate of 1e-3.

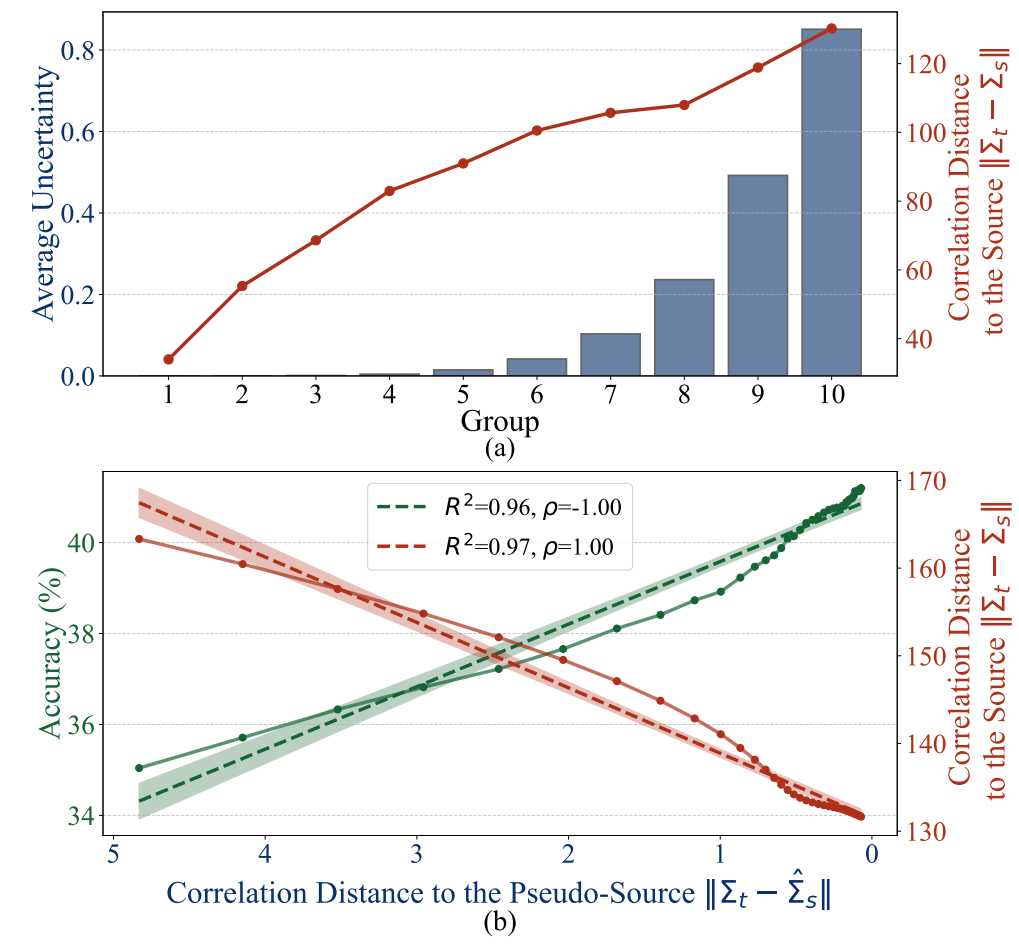

*Figure 3.* Experimental validation of theories. (a) Average uncertainty and correlation distance to source domain of each group, groups with lower uncertainty exhibit smaller correlation distances. (b) Relationships between ACC, correlation distance to the source, and correlation distance to the pseudo-source, both ACC and $\|\Sigma_t - \Sigma_s\|$ are strongly linearly related to $\|\Sigma_t - \hat{\Sigma}_s\|$.

## 5. Experiments

### 5.1. Experimental settings

Following previous studies, we evaluate the adaptation performance on two main tasks: domain generalization (`PACS` (Li et al., 2017), `OfficeHome` (Venkateswara et al., 2017), and `DomainNet` (Peng et al., 2019) dataset) and image corruption (`CIFAR-10-C`,`CIFAR-100-C`, and `ImageNet-C` (Hendrycks & Dietterich, 2019)). What's more, we also evaluate our method on multimodal tasks based on CLIP (Radford et al., 2021). The comparison methods include `backpropagation-free` (BN (Schneider et al., 2020), T3A (Iwasawa & Matsuo, 2021), AdaNPC (Zhang et al., 2023)) and `backpropagation-based` methods (TENT (Wang et al., 2020), PLC (Lee, 2013), EATA (Niu et al., 2022), SAR (Niu et al., 2023), TSD (Wang et al., 2023), TIPI (Nguyen et al., 2023), TEA (Yuan et al., 2024)). Backbone networks include `ResNet-18/50` (He et al., 2016) and `ViT-B/16` (Dosovitskiy, 2020). Additionally, the evaluation encompasses multiple aspects, including accuracy, efficiency, and resistance to forgetting. For LinearTCA⁺, we report its results combined with the best baseline. Refer to Appendix D for more implement information. For further experimental results and analysis, please see Appendix E.

### 5.2. Experimental validation of theories

**For Theorem 3.5:** Correlation of high-certainty test instances approximates the source correlation. We divide the test embeddings of `CIFAR-10-C` under `ResNet-18` into 10 groups based on prediction uncertainty and calcu-

| Domain | Method | PACS | | | | OfficeHome | | | | DomainNet | | | | AVG |
|---|---|---|---|---|---|---|---|---|---|---|---|---|---|---|
| | | ResNet-18 | ResNet-50 | ViT-B/16 | AVG | ResNet-18 | ResNet-50 | ViT-B/16 | AVG | ResNet-18 | ResNet-50 | ViT-B/16 | AVG | |
| | SOURCE | 81.84 | 84.78 | 87.02 | 84.54 | 62.01 | 67.01 | 76.11 | 68.37 | 39.13 | 43.58 | 50.29 | 44.33 | 65.75 |
| BP-Free | BN | 82.65 | 84.99 | - | - | 62.05 | 66.30 | - | - | 37.93 | 41.94 | - | - | - |
| | T3A | 83.20 | 85.71 | 88.06 | 85.66 | 63.26 | 67.85 | 78.87 | 69.99 | 40.62 | 44.92 | 53.94 | 46.49 | 67.38 |
| | AdaNPC | 83.48 | 86.12 | 89.11 | 86.24 | 62.88 | 67.05 | 77.26 | 69.07 | 40.50 | 45.17 | 53.28 | 46.32 | 67.21 |
| BP-Based | TENT | 85.23 | 88.07 | 84.98 | 86.09 | 63.09 | 67.67 | 76.95 | 69.24 | 39.42 | 43.97 | 39.96 | 41.12 | 65.48 |
| | PLC | 83.16 | 86.59 | 87.97 | 85.91 | 62.22 | 66.44 | 76.51 | 68.39 | 37.96 | 41.63 | 47.29 | 42.30 | 65.53 |
| | EATA | 83.30 | 84.68 | 86.60 | 84.86 | 62.49 | 67.01 | 76.98 | 68.83 | 41.65 | 46.89 | 54.40 | 47.65 | 67.11 |
| | SAR | 85.41 | 85.79 | 87.12 | 86.11 | 62.51 | 67.94 | 76.66 | 69.04 | 38.49 | 42.19 | 42.81 | 41.16 | 65.44 |
| | TIPI | 87.39 | 88.01 | 87.98 | 87.79 | 63.25 | 68.36 | 77.09 | 69.57 | 36.05 | 44.08 | 39.70 | 39.94 | 65.77 |
| | TEA | 87.19 | 88.75 | 87.37 | 87.77 | 63.43 | 68.56 | 76.15 | 69.38 | 39.43 | 43.48 | 48.41 | 43.78 | 66.98 |
| | TSD | 87.83 | 89.99 | 83.43 | 87.08 | 62.47 | 68.63 | 75.49 | 68.87 | 38.59 | 42.12 | 48.72 | 43.14 | 66.36 |
| Ours | LinearTCA | 83.59 | 86.78 | 88.61 | 86.33 | 63.66 | 68.43 | 78.26 | 70.12 | 40.79 | 44.89 | 52.79 | 46.16 | 67.53 |
| | LinearTCA$^{+}$ | **88.77** | **90.68** | **89.30** | **89.58** | **64.27** | **69.32** | **79.02** | **70.87** | **42.20** | **47.17** | **55.49** | **48.29** | **69.58** |

| ImgCop | Method | CIFAR-10-C | | | | CIFAR-100-C | | | | ImageNet-C | | | | AVG |
|---|---|---|---|---|---|---|---|---|---|---|---|---|---|---|
| | | ResNet-18 | ResNet-50 | ViT-B/16 | AVG | ResNet-18 | ResNet-50 | ViT-B/16 | AVG | ResNet-18 | ResNet-50 | ViT-B/16 | AVG | |
| | SOURCE | 50.80 | 50.77 | 71.48 | 57.68 | 31.01 | 34.02 | 51.71 | 38.91 | 14.70 | 18.15 | 39.83 | 24.23 | 40.27 |
| BP-Free | BN | 73.70 | 72.24 | - | - | 48.38 | 48.41 | - | - | 27.59 | 32.06 | - | - | - |
| | T3A | 58.89 | 54.87 | 74.21 | 62.65 | 32.52 | 34.94 | 54.24 | 40.57 | 14.56 | 18.05 | 39.78 | 24.13 | 42.45 |
| | AdaNPC | 57.72 | 54.75 | 74.60 | 62.36 | 29.70 | 32.27 | 53.21 | 38.39 | 11.93 | 15.62 | 36.78 | 21.44 | 40.73 |
| BP-Based | TENT | 75.21 | 72.33 | 71.48 | 73.01 | 50.82 | 50.12 | 52.72 | 51.22 | 35.39 | 41.32 | 48.01 | 41.57 | 55.27 |
| | PLC | 73.72 | 72.34 | 71.46 | 72.51 | 48.35 | 48.38 | 51.71 | 49.48 | 27.59 | 32.06 | 38.74 | 32.80 | 51.59 |
| | EATA | 73.86 | 72.38 | 73.67 | 73.30 | 49.71 | 49.89 | 62.40 | 54.00 | 39.19 | 48.17 | 64.36 | 50.58 | 59.29 |
| | SAR | 73.97 | 73.37 | 71.48 | 72.94 | 51.60 | 50.25 | 54.29 | 52.05 | 38.55 | 46.30 | 57.94 | 47.60 | 57.53 |
| | TIPI | 76.10 | 72.46 | 71.48 | 73.35 | 50.61 | 50.30 | 52.36 | 51.09 | 35.73 | 41.87 | 48.50 | 42.03 | 55.49 |
| | TEA | 76.20 | 72.54 | 71.48 | 73.41 | 50.67 | 50.21 | 52.31 | 51.06 | 32.38 | 38.90 | 41.37 | 37.55 | 54.01 |
| | TSD | 76.93 | 73.23 | 71.47 | 73.88 | 49.35 | 49.60 | 51.74 | 50.23 | 30.11 | 35.08 | 41.33 | 35.51 | 53.20 |
| Ours | LinearTCA | 60.96 | 60.27 | 77.26 | 66.16 | 35.03 | 37.28 | 55.42 | 42.58 | 16.07 | 19.34 | 41.37 | 25.60 | 44.78 |
| | LinearTCA$^{+}$ | **77.13** | **73.53** | **79.55** | **76.74** | **52.08** | **51.17** | **63.71** | **55.65** | **39.21** | **48.22** | **64.71** | **50.71** | **61.04** |

*Table 1.* Accuracy comparison of different TTA methods based on `ResNet-18/50` and `ViT-B/16` backbones. The upper part of the table corresponds to the domain generalization task, while the lower part corresponds to the image corruption task. The best results are highlighted in **boldface**, and the second ones are underlined. "-" indicates that `ViT-B/16` does not include any BN layers.

late the correlation distance between each group and the original source. As shown in Figure 3a, groups with lower uncertainty exhibit smaller correlation distances, indicating a closer approximation to the source correlation.

**For Theorem 3.6 and Corollary 3.7:** Test-time correlation alignment reduces test classification error. We iteratively optimize $W$ and record the correlation distances between test domain and pseudo-source domain, $\|\Sigma_t - \hat{\Sigma}_s\|$, as well as the true distances between test domain and source domain, $\|\Sigma_t - \Sigma_s\|$, and ACC. As shown in Figure 3b, under a linear fit ($R^2 = 0.97$), $\|\Sigma_t - \hat{\Sigma}_s\|$ is strongly positively related to $\|\Sigma_t - \Sigma_s\|$ (Spearman correlation coefficient = 1). Under $R^2 = 0.96$, it is strongly negatively related to ACC (Spearman correlation coefficient = -1). This further validates that pseudo-source correlation alignment promotes alignment with the original source. Additionally, pseudo-source correlation alignment effectively reduces test classification error, thus improving the model's domain adaptation capability.

### 5.3. Comparison with TTA Methods

**Accuracy.** Table 1 presents ACC comparisons between TCA methods and state-of-the-art TTA approaches across various benchmarks, backbones, and tasks. (1) As a plug-and-play module, LinearTCA$^{+}$ consistently enhances performance across all datasets and backbones, achieving a new state-of-the-art. Notably, on the `CIFAR-10-C` dataset with the `ViT-B/16` backbone, LinearTCA$^{+}$ shows substantial improvements over the best-performing baseline, with an increase of 4.95%. (2) Across datasets, LinearTCA shows robust improvement compared to the source model, with average gains of 1.79%, 1.75%, 1.78%, 8.48%, 3.67% and 4.51%, respectively. Particularly, on the `OfficeHome` and `DomainNet` dataset, LinearTCA outperforms all baseline methods. However, on datasets such as `CIFAR-10/100-C` and `ImageNet-C`, although LinearTCA yields ACC gains of 8.48%, 3.67% and 4.51% over the source model, it falls short of some advanced methods. (3) Across backbones, LinearTCA also demonstrates robust improvements compared to the source model, especially with the `ViT-B/16` backbone, surpassing the highest-performing baseline on most datasets. We provide a detailed analysis of these experimental results in Section 5.5 to further reveal the strengths and limitations of LinearTCA.

| Type | Method | Memory(MB) | | | | Time(s) | | | |
|---|---|---|---|---|---|---|---|---|---|
| | | ResNet-18 | ResNet-50 | ViT-B/16 | AVG | ResNet-18 | ResNet-50 | ViT-B/16 | AVG |
| | SOURCE | 920.61 | 878.87 | 917.02 | 905.50 | 3.92 | 9.16 | 3.98 | 5.69 |
| BP-Free | BN | +0.25 | +48.57 | - | - | +0.88 | +4.80 | - | - |
| | T3A | +1.00 | +4.43 | +2.02 | +2.48 | +1.98 | +3.62 | +12.22 | +5.94 |
| | AdaNPC | +2.04 | +8.23 | +2.96 | +4.41 | +1.73 | +2.78 | +12.08 | +5.53 |
| BP-Based | TENT | +1883.63 | +4788.93 | +5246.53 | +3973.03 | +3.85 | +11.52 | +12.27 | +9.22 |
| | PLC | +1934.14 | +4787.26 | +8624.95 | +5115.45 | +5.94 | +9.51 | +25.86 | +13.77 |
| | EATA | +5332.44 | +10838.53 | +11172.56 | +9114.51 | +1.76 | +4.20 | +22.81 | +9.59 |
| | SAR | +2642.82 | +5380.18 | +5401.31 | +4474.77 | +11.23 | +23.31 | +54.08 | +29.54 |
| | TSD | +2025.07 | +5162.55 | +9280.69 | +5489.44 | +4.70 | +13.47 | +34.68 | +17.62 |
| | TEA | +7316.95 | +15733.10 | +16082.00 | +13044.02 | +123.14 | +278.87 | +596.28 | +332.76 |
| | TIPI | +2520.01 | +10660.83 | +12542.71 | +8574.52 | +26.54 | +49.73 | +45.25 | +40.51 |
| Ours | **TCA** | **+0.00** | **+0.00** | **+0.00** | **+0.00** | **+0.06** | **+0.07** | **+0.08** | **+0.07** |

*Table 2.* Maximum GPU memory usage and running time of different TTA methods on `CIFAR-10-C`.

**Efficiency.** We evaluate each method's efficiency in terms of peak GPU memory usage and total runtime. Table 2 reports results on the `CIFAR-10-C` dataset across different

| Type | Method | PACS | OfficeHome | CIFAR-10-C | CIFAR-100-C | AVG |
|---|---|---|---|---|---|---|
| BP-Free | SOURCE | 99.35 | 94.40 | 92.36 | 70.39 | 89.12 |
| | BN | 98.90 (-0.44) | 92.85 (-1.55) | 62.98 (-29.38) | 39.45 (-30.94) | 73.55 (-15.58) |
| | T3A | 99.33 (-0.01) | 93.31 (-1.09) | 91.95 (-0.41) | 65.66 (-4.73) | 87.56 (-1.56) |
| | AdaNPC | 99.28 (-0.06) | 93.31 (-1.09) | **92.00 (-0.36)** | 63.88 (-6.51) | 87.12 (-2.01) |
| | LinearTCA | **99.42 (+0.08)** | **93.87 (-0.53)** | 91.16 (-1.20) | **67.35 (-3.04)** | **87.95 (-1.17)** |
| | LinearTCA w/o $W$ | **99.35 (0.00)** | **94.40 (0.00)** | **92.36 (0.00)** | **70.39 (0.00)** | **89.12 (0.00)** |
| BP-Based | TENT | 96.74 (-2.61) | 92.79 (-1.61) | 90.26 (-2.10) | 67.27 (-3.12) | 86.76 (-2.36) |
| | PLC | 97.12 (-2.23) | 92.73 (-1.67) | 63.05 (-29.31) | 39.48 (-30.91) | 73.09 (-16.03) |
| | EATA | 98.33 (-1.02) | **93.66 (-0.74)** | 90.24 (-2.12) | 68.52 (-1.87) | 87.69 (-1.44) |
| | SAR | 97.12 (-2.23) | 86.35 (-8.05) | 90.31 (-2.05) | 68.77 (-1.62) | 85.63 (-3.49) |
| | TSD | 95.10 (-4.24) | 85.37 (-9.03) | 67.78 (-24.58) | 39.48 (-30.91) | 71.93 (-17.19) |
| | TEA | 90.22 (-9.13) | 93.30 (-1.10) | 90.60 (-1.76) | 68.93 (-1.46) | 85.76 (-3.36) |
| | TIPI | 98.15 (-1.20) | 92.79 (-1.61) | 70.75 (-21.61) | 46.03 (-24.36) | 76.93 (-12.20) |
| | LinearTCA$^+$ | **99.03 (-0.31)** | 93.65 (-0.75) | **90.68 (-1.68)** | **69.05 (-1.34)** | **88.10 (-1.02)** |

*Table 4.* The accuracy of different TTA methods when returning to the source domain after adaptation. "BP-Free" indicates backpropagation-free TTA methods, while "BP-Based" denotes backpropagation-dependent ones.

backbones. TCA consistently achieves the lowest memory and time cost. For memory, since we record peak memory consumption, LinearTCA exhibits minimal independent memory usage (as shown in Table 3) and thus does not impose additional memory constraints on the device. In contrast, other methods are embedded within the model's forward and backward propagation processes, significantly increasing peak memory usage (e.g., TEA uses 15× the memory of the source model). For runtime, with a `ViT-B/16` backbone, LinearTCA requires only 0.6% of AdaNPC's time. These results highlight LinearTCA's high efficiency, making it well-suited for resource-constrained edge deployment.

| Method | ResNet18 | ResNet50 | ViT-B/16 | AVG |
|---|---|---|---|---|
| LinearTCA | 118.56 | 448.64 | 452.11 | 339.77 |

*Table 3.* Independent maximum GPU memory usage of LinearTCA on CIFAR-10-C.

**Forgetting resistance.** Table 4 shows the change in accuracy when each method (using `ResNet-18`) returns to the source domain after adaptation. "LinearTCA w/o $W$" refers to the variant without the linear transformation, which is equivalent to the original source model and thus retains full source knowledge. Despite applying $W$, LinearTCA demonstrates much stronger resistance to forgetting than other methods—especially on PACS, where it even improves source performance, showing positive backward transfer. Moreover, LinearTCA$^+$ further enhances the forgetting robustness of existing TTA methods.

### 5.4. Performance on Closed-Source Foundation Models

To validate TCA's effectiveness on closed-source foundation models, we conduct experiments with CLIP (Radford et al., 2021) on PACS, OfficeHome, and VLCS datasets, following the experimental setup in WATT (Osowiechi et al., 2024). As shown in Table 5, TCA achieving performance improvements of 1.28%, 2.08%, and 2.85% on the three datasets respectively. The superior results stem from our method's explicit alignment of embedding distributions with the source domain, which proves particularly effective for multi-modal models like CLIP that compute image-text similarity directly. While LinearTCA$^+$ holds a slight advantage,

| Method | PACS A | PACS C | PACS P | PACS S | AVG |
|---|---|---|---|---|---|
| CLIP† | 97.44 | 97.38 | 99.58 | 86.06 | 95.12 |
| TENT† | 97.54±0.02 | 97.37±0.04 | 99.58±0.00 | 86.37±0.05 | 95.22 |
| TPT† | 95.10±0.41 | 91.42±0.22 | 98.56±0.40 | 87.23±0.06 | 93.08 |
| CLIPArTT† | 97.64±0.02 | 97.37±0.02 | 99.58±0.00 | 86.79±0.04 | 95.35 |
| WATT-P† | 97.49±0.08 | 97.47±0.04 | 99.58±0.00 | 89.73±0.16 | 96.07 |
| WATT-S† | 97.66±0.08 | 97.51±0.02 | 99.58±0.00 | 89.56±0.14 | 96.08 |
| LinearTCA | 97.80 | **99.39** | **99.94** | 92.32 | **97.36** |
| LinearTCA$^+$ | **97.87±0.06** | 99.20±0.02 | **99.94±0.00** | **92.36±0.06** | 97.34 |

| Method | OfficeHome A | OfficeHome C | OfficeHome P | OfficeHome R | AVG |
|---|---|---|---|---|---|
| CLIP† | 79.30 | 65.15 | 87.34 | 89.31 | 80.28 |
| TENT† | 79.26±0.14 | 65.64±0.05 | 87.49±0.02 | 89.50±0.04 | 80.47 |
| TPT† | 81.97±0.17 | 67.01±0.21 | 89.00±0.06 | 89.66±0.06 | 81.91 |
| CLIPArTT† | 79.34±0.05 | 65.69±0.11 | 87.35±0.07 | 89.29±0.03 | 80.42 |
| WATT-P† | 80.37±0.25 | 68.59±0.13 | 88.15±0.07 | 90.18±0.03 | 81.82 |
| WATT-S† | 80.43±0.09 | 68.26±0.11 | 88.02±0.08 | 90.14±0.06 | 81.71 |
| LinearTCA | 85.55 | 68.70 | 90.26 | **90.58** | 83.77 |
| LinearTCA$^+$ | **85.62±0.38** | **69.25±0.1** | **90.29±0.01** | 90.42±0.1 | **83.90** |

| Method | VLCS C | VLCS L | VLCS S | VLCS V | AVG |
|---|---|---|---|---|---|
| CLIP† | 99.43 | 67.75 | 71.74 | 84.90 | 80.96 |
| TENT† | 99.43±0.00 | 67.31±0.14 | 71.57±0.15 | **85.10±0.11** | 80.85 |
| TPT† | 97.62±0.12 | 49.77±0.03 | 71.56±0.86 | 71.17±0.70 | 72.53 |
| CLIPArTT† | 99.43±0.00 | 67.74±0.10 | 71.67±0.01 | 84.73±0.08 | 80.89 |
| WATT-P† | 99.36±0.00 | 67.55±0.39 | 74.75±0.07 | 82.53±0.10 | 81.05 |
| WATT-S† | 99.36±0.00 | 68.59±0.25 | 75.16±0.12 | 83.24±0.05 | 81.59 |
| LinearTCA | 99.86 | 73.98 | 78.47 | 84.41 | 84.18 |
| LinearTCA$^+$ | **99.88±0.03** | **74.39±0.1** | **79.44±0.22** | 84.06±0.14 | **84.44** |

*Table 5.* The accuracy comparison of different methods on PACS, OfficeHome, and VLCS datasets using CLIP-ViT-B/16. †: numbers are from WATT (Osowiechi et al., 2024). The best results are highlighted in **boldface**, and the second ones are underlined.

both variants perform similarly, suggesting that even simple correlation alignment can notably enhance performance on popular models like CLIP. This underscores its effectiveness as a versatile plug-and-play module for improving diverse adaptation methods.

### 5.5. Analysis

**Effective range of LinearTCA.** As discussed in Section 5.3, although LinearTCA$^+$ significantly improves all TTA methods, LinearTCA only achieves SOTA performance on part of datasets and backbones. The reasons may be: 1) Although the highest-certainty embeddings are selected as pseudo-source domains, if these embeddings still exhibit substantial differences from the true source domain (or if the backbone's feature extraction capacity is insufficient, e.g., `ResNet-18` vs. `ViT-B/16`), the performance ceiling of LinearTCA is limited. In contrast, other TTA methods update the model, thereby raising this ceiling and facilitating easier correlation alignment for LinearTCA$^+$. 2) We only use a linear transformation $W$ for alignment, which may work well for simple shifts; however, the true distribution shifts may not conform to linear transformations but exhibit complex nonlinear relationships. We design a demo experiment to validate this hypothesis. In Figure 4a and b, the test domain shifts are linear and nonlinear, respectively. As shown, the transformed embeddings in Figure 4a align well with the original distribution, while the perfor-

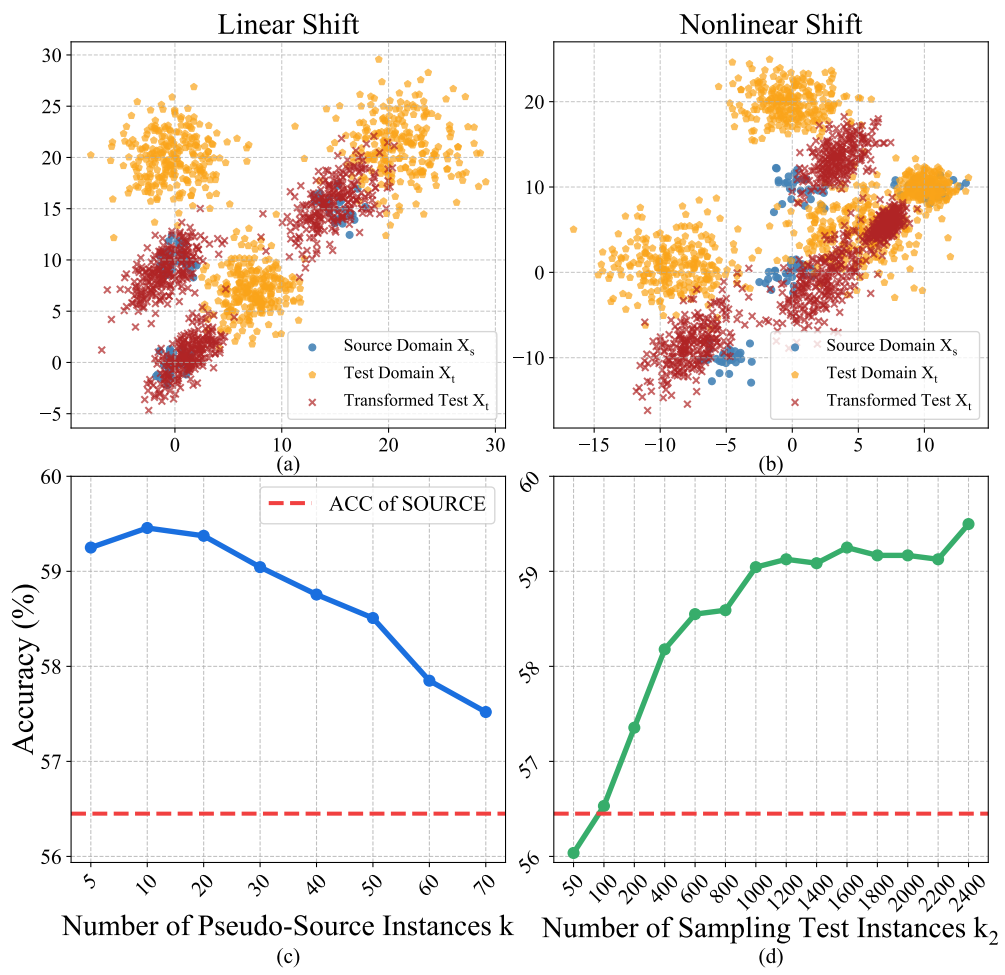

*Figure 4.* Analysis of TCA. (a) When the test domain (yellow) undergoes a nearly linear shift from the source domain (blue), after adaptation by LinearTCA, the transformed test domain (red) is well-aligned with the source. (b) In the case of a nonlinear shift, although partial alignment is achieved, it is still insufficient. (c) and (d) Ablation study examining the effect of pseudo-source domain size and test domain size.

| Backbone | Method | PACS | | | | AVG | OfficeHome | | | | AVG |
|---|---|---|---|---|---|---|---|---|---|---|---|
| | | A | C | P | S | | A | C | P | R | |
| ResNet-18 | Source | 78.37 | 77.39 | 95.03 | 76.58 | 81.84 | 56.45 | 48.02 | 71.34 | 72.23 | 62.01 |
| | LinearTCA | 80.91 | 81.02 | 95.69 | 76.74 | 83.59 | 59.46 | 50.40 | 72.02 | 72.78 | 63.66 |
| | LinearTCA$^+$ | 88.38 | 87.12 | 96.59 | 83.00 | 88.77 | 59.83 | 51.80 | 72.29 | 73.17 | 64.27 |
| | LinearTCA(MLP-2) | 81.24 | 81.73 | 95.89 | 78.15 | 84.25 | 59.62 | 50.84 | 72.07 | 72.94 | 63.87 |
| | LinearTCA$^+$(MLP-2) | **88.68** | 87.15 | **96.68** | 83.19 | 88.93 | **59.83** | 51.80 | **72.79** | 73.46 | 64.47 |
| | LinearTCA(MLP-3) | 81.62 | 81.81 | 96.03 | 79.35 | 84.70 | 59.62 | 50.65 | 72.07 | 73.02 | 63.84 |
| | LinearTCA$^+$(MLP-3) | 88.38 | **87.23** | 96.59 | **83.36** | **88.98** | **59.83** | **52.08** | **72.79** | **73.54** | **64.56** |

*Table 6.* Extending LinearTCA/LinearTCA$^+$ by introducing MLP-based transformations with two (MLP-2) and three (MLP-3) layers The best results are highlighted in **boldface**, and the second ones are underlined.

mance in Figure 4b shows partial alignment which is still insufficient. We further explore the utilization of nonlinear architecture (MLP) for calculating transformation $W$. As shown in Table 6, incorporating nonlinear activations with deeper architectures leads to further improvements.

**Ablation study.** Our method involves only one hyperparameter—the number of pseudo-source embeddings $k$. Since the total number of test samples is often unknown in practice, we also sample $k_2$ embeddings from the test set to study its impact. As shown in Figure 4c,d, LinearTCA achieves the best accuracy on `OfficeHome` when $k = 10$ and $k_2 = 2400$. Importantly, it consistently outperforms the source model across a wide range of $k$ and $k_2$, demonstrating strong practical applicability.

**Upper performance bound for TCA.** To assess the upper bound of TCA, we conduct two additional experiments in Table 7: (a) fine-tuning directly on the target domain; (b) applying LinearTCA and LinearTCA$^+$ with real source distributions. Compared to the original LinearTCA$^+$, approach (b) further improves performance, by 0.38% on PACS and 1.03% on OfficeHome. Both (a) and (b) outperform the original LinearTCA in most domains. On OfficeHome, even the simpler LinearTCA with real source data (b) surpasses fine-tuning (a), highlighting the importance of source distribution and the effectiveness of approximating it in TCA.

| Backbone | Method | PACS | | | | AVG | OfficeHome | | | | AVG |
|---|---|---|---|---|---|---|---|---|---|---|---|
| | | A | C | P | S | | A | C | P | R | |
| ResNet-18 | Source | 78.37 | 77.39 | 95.03 | 76.58 | 81.84 | 56.45 | 48.02 | 71.34 | 72.23 | 62.01 |
| | LinearTCA | 80.91 | 81.02 | 95.69 | 76.74 | 83.59 | 59.04 | 49.97 | 71.77 | 72.89 | 63.42 |
| | LinearTCA$^+$ | 88.38 | 87.12 | 96.59 | 83.00 | 88.77 | 59.33 | 51.18 | 72.20 | 71.72 | 63.61 |
| | TCA(a) | 86.18 | 82.67 | 95.03 | 80.81 | 86.17 | 58.69 | 50.80 | 72.04 | 72.92 | 63.61 |
| | LinearTCA(b) | 81.59 | 81.48 | 96.05 | 77.51 | 84.15 | 59.94 | 51.63 | 72.36 | 73.48 | 64.35 |
| | LinearTCA$^+$(b) | 88.98 | 87.57 | 96.74 | 83.30 | 89.15 | 60.03 | 52.29 | 72.55 | 73.87 | 64.64 |

*Table 7.* Upper performance bound for TCA. TCA(a): Fine-tuning directly on the target distribution. LinearTCA(b) and LinearTCA$^+$(b): Applying LinearTCA and LinearTCA$^+$ with real source distributions.

| Method | Art Domian of OfficeHome | | | | | | | | | | | AVG |
|---|---|---|---|---|---|---|---|---|---|---|---|---|
| Batch Size | 1 | 2 | 4 | 8 | 16 | 32 | 64 | 128 | 256 | 512 | 1024 | |
| Estimation error | 2542 | 2414 | 2434 | 2430 | 2415 | 2417 | 2437 | 2415 | 2413 | 2424 | 2427 | 2433 |
| Source | 56.45 | 56.45 | 56.45 | 56.45 | 56.45 | 56.45 | 56.45 | 56.45 | 56.45 | 56.45 | 56.45 | 56.45 |
| TEA | 0.824 | 18.01 | 40.79 | 49.23 | 55.54 | 55.71 | 57.35 | 58.55 | 57.11 | 57.82 | 57.93 | 46.26 |
| LinearTCA | **58.61** | **58.61** | **58.57** | **58.77** | **58.94** | **58.86** | 59.06 | 59.46 | 59.27 | 59.35 | 59.56 | **59.05** |
| LinearTCA$^+$ | 0.824 | 18.54 | 41.37 | 51.13 | 56.05 | 58.44 | **59.3** | **59.83** | **59.66** | **59.86** | **59.96** | 47.72 |

*Table 8.* Accuracy comparisons of different TTA methods on the Art domain of OfficeHome dataset with varying batch sizes based on ResNet-18. The best results are highlighted in **boldface**, and the second ones are underlined.

**Performance under difference batch sizes.** To study the impact of batch size, we evaluate TCA's performance and pseudo-source estimation error under varying batch sizes in Table 8. Even with batch size 1, LinearTCA outperforms the source model by 2.16%, and LinearTCA$^+$ consistently improves over TEA across all settings. This robustness stems from TCA's incremental estimation of test-domain covariance, which converges over time. While small batch sizes mainly affect early predictions, their influence diminishes as more data is seen. Moreover, the pseudo-source estimation error remains unaffected by batch size, since it relies on a small set of high-confidence samples (Figure 4c) and benefits from the same incremental computation.

## 6. Conclusion and Future Work

In this paper, we introduce the Test-time Correlation Alignment (TCA) to address the chanllenges in Test-Time Adaptation (TTA), such as overlooking feature correlation, overhead computation and domain forgetting. TCA is a novel paradigm that enhances test-time adaptation (TTA) by aligning the correlation of high-certainty instances and test instances and is demonstrated with a theoretical guarantee. Extensive experiments validate our theoretical insights and show that TCA methods significantly outperforms baselines on accuracy, efficiency, and forgetting resistance across various tasks, benchmarks and backbones.

Future work may incorporate more nonlinear transformations for more effective TCA. Additionally, with the interesting "positive backward transfer" phenomenon in Table 4, we will further investigate the underlying mechanism.

## Impact Statement

This paper presents work whose goal is to advance the field of Machine Learning. There are many potential societal consequences of our work, none which we feel must be specifically highlighted here.

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

# Test-time Correlation Alignment

## Appendix

The structure of Appendix is as follows:

- Appendix A contains the extended related work.
- Appendix B contains all missing proofs in the main manuscript.
- Appendix C details the proposed methods LinearTCA and LinearTCA$^+$.
- Appendix D details the dataset and implementation.
- Appendix E contains additional experimental results.

## A. Extended Related Work

### A.1. Correlation Alignment

Correlation alignment is a crucial technique in unsupervised domain adaptation (UDA) designed to address domain shift problems. In real-world scenarios, significant domain shifts often occur between training and test data, which can severely degrade the performance of conventional machine learning methods. To tackle this challenge, CORrelation ALignment (CORAL) (Cheng et al., 2021a) is introduced to align the feature-wise statistics of the source and target distributions through a linear transformation. Similar to CORAL, Maximum Mean Discrepancy (MMD) (Gretton et al., 2006) is another technique for mitigating domain gap by minimizing the mean discrepancy between different domains. Unlike CORAL, which focuses on feature-wise correlations, MMD match the instance-wise statistics of the domain distribution.

Correlation Alignment has been extended and applied in several innovative ways. DeepCORAL (Sun & Saenko, 2016) extends CORAL to deep neural networks by employing a differentiable Correlation Alignment loss function. This enables end-to-end domain adaptation and facilitates more effective nonlinear transformations, thereby enhancing generalization performance on unsupervised target domains. DeerCORAL (Das et al., 2021) leverages CORAL loss in combination with synthetic data to address long-tailed distributions in real-world scenarios. High-order CORAL (Cheng et al., 2021b), which is inspired by MMD and CORAL, utilizes third-order correlation to capture more detailed statistical information and effectively characterize complex, non-Gaussian distributions. IJDA (Qian et al., 2023) introduces a novel metric that combines MMD and CORAL to improve distribution alignment and enhance domain confusion.

In addition to these advancements, recent studies have explored the integration of CORAL into more complex models and settings. For example, CAADG (Rahman et al., 2020a) presents a domain generalization framework that combines CORAL with adversarial learning to jointly adapt features and minimize the domain disparity. Moreover, JCGNN (Wang et al., 2021) integrates CORAL into Graph Neural Network (GNN) to generate the domain-invariant features.

Although CORAL has achieved significant success in domain adaptation (DA), its application in test-time adaptation (TTA) is constrained by privacy and resource limitations, which make it infeasible to compute the source correlation. This limitation significantly hampers the practicality of CORAL in more real-world scenarios, such as test-time correlation alignment (TCA).

### A.2. Test-Time Adaptation

In real-world scenarios, test data often undergoes natural variations or corruptions, leading to distribution shifts between the training and testing domains. Recently, various Test-Time Adaptation (TTA) approaches have been proposed to adapt pre-trained models during testing. These methods can be broadly categorized into batch normalization calibration methods, pseudo-labeling methods, consistency training methods, and clustering-based training methods (Liang et al., 2024). For further discussion, we classify them into two groups based on their dependence on backpropagation, as outlined in (Niu et al., 2024).

Backpropagation (BP)-Free TTA: This group includes batch normalization (BN) calibration methods (Wu et al., 2024; Schneider et al., 2020) and certain pseudo-labeling methods (Zhang et al., 2023) that do not update model parameters. BN-based methods posit that the statistics in BN layers capture domain-specific knowledge. To mitigate the domain gap, these methods replace training BN statistics with updated statistics computed from the target domain. Some pseudo-labeling methods such as T3A (Iwasawa & Matsuo, 2021) utilize prototype similarity and AdaNPC (Zhang et al., 2023) utilize k-nearest neighbor (kNN) to refine predictions. Although BP-Free TTA methods are computationally efficient, their image corruption adaptation capabilities are often limited.

Backpropagation (BP)-Based TTA: This group encompasses certain pseudo-labeling methods (Zeng et al., 2024), consistency training methods (Sinha et al., 2023), and clustering-based training methods (Lee et al., 2024). Some pseudo-labeling methods use filtering strategies, such as thresholding or entropy-based approaches, to generate reliable pseudo-labels, thereby reducing the discrepancy between predicted and pseudo-labels. For instance, PLC (Lee, 2013) updates classifier layer parameters with certain pseudo-labels during adaptation. TSD (Wang et al., 2023) filters unreliable features or predictions with high entropy, as lower entropy correlates with higher accuracy, and applies a consistency filter to refine instances further. Consistency training methods aim to enhance the stability of network predictions or features by addressing variations in input data, such as noise or perturbations, and changes in model parameters. TIPI (Nguyen et al., 2023), for example, simulates domain shifts via input transformations and employs regularizers to maintain model invariance. Clustering-based training methods leverage clustering techniques to group target features, and reduce uncertainty in predictions and improving model robustness. TENT (Wang et al., 2020) minimizes prediction entropy on target data, while EATA (Niu et al., 2022) selects reliable instances to minimize entropy loss and applies a Fisher regularizer. SAR (Niu et al., 2023) removes noisy instances with large gradients and encourages model weights to converge toward a flat minimum, enhancing robustness against residual noise. Generally, BP-Based TTA methods demonstrate superior domain adaptation capabilities compared to BP-Free methods, but they typically require multiple backward propagations for each test instance, leading to computational inefficiencies.

Despite their strengths, both BP-Free and BP-Based TTA methods perform instance-wise alignment without considering feature correlation alignment. Our proposed method, TCA, is orthogonal to most existing TTA methods. It achieves both instance-wise and correlation alignment without backpropagation. TCA is a theoretically supported TTA paradigm that effectively addresses the challenges of efficiency and domain forgetting. By applying a simple linear transformation, TCA performs both instance and correlation alignment without requiring additional model updates. Moreover, it can function as a plug-and-play module to enhance the performance of existing TTA methods.

# B. Proof of Theoretical Statement

## B.1. Proof of Theorem 3.5

Here, we present Theorem 3.5 again for convenience.

**Theorem 3.5** Let $h_\theta(\cdot) = g(f(\cdot))$ be an L-Lipschitz continuous hypothesis on $\mathcal{H}$. $\Omega := \bigcup_{x \in \mathbb{D}_t} \mathcal{B}(x, r^*)$ is the set of balls near the test data. We sample $k$ source instances from $\mathbb{D}_s \cap \Omega$ and $k$ test instances from $\mathbb{D}_t$ to obtain $[X_s, Z_s, P_s]$ and $[X_t, Z_t, P_t]$ by $h_\theta(\cdot)$, respectively. Per Assumption 3.2, Assumption 3.3 and Assumption 3.4, with a probability of at least$1 - \exp\left(-\frac{(c_t \mu^- \pi_{d_I} r^{d_I} n_s - 1)^2}{2 c_t \mu^- \pi_{d_I} r^{d_I} n_s} + \log k\right)$, we have

$$\|Z_t - Z_s\| \leq \frac{\|P_t - P_s\| + \|o(kr^*)\|}{\|J_g(Z_s)\|} \tag{22}$$

where $\pi_{d_I} = \lambda(\mathcal{B}(0,1))$ is the volume of the $d_I$ dimension unit ball and $d_I$ is the dimension of input $x$. Furthermore, considering the true source correlation $\Sigma_s = \mathbb{E}[\tilde{Z}_s^T \tilde{Z}_s]$ and the pseudo-source correlation $\hat{\Sigma}_s = \tilde{Z}_t^T \tilde{Z}_t$, where $\tilde{Z}_s$ and $\tilde{Z}_t$ are centered. With a probability of at least $\min(1 - \exp\left(-\frac{(c_t \mu^- \pi_{d_I} r^{d_I} n_s - 1)^2}{2 c_t \mu^- \pi_{d_I} r^{d_I} n_s} + \log k\right), 1 - \delta)$, the correlation

distance $\|\Sigma_s - \hat{\Sigma}_s\|$ is bounded by:

$$\|\Sigma_s - \hat{\Sigma}_s\|_F \leq$$
$$2\|Z_s\|_F(\frac{\|\hat{Y}_t - P_t\|_F + A}{\|J_g(Z_s)\|_F}) + (\frac{\|\hat{Y}_t - P_t\|_F + A}{\|J_g(Z_s)\|_F})^2 + B \tag{23}$$

where $\hat{Y}_t$ is the one-hot encoding of $P_t$, $A = \|o(kr^*)\| + k\epsilon(h_\theta(X_t)) + k\epsilon(h_\theta(X_s))$ represents the output error of the sampled instances, and $B = \sqrt{\frac{\log(2/\delta)}{2k}}$ is the sampling error.

We begin by proving Equation (22). According to Assumption 3.3 and Assumption 3.4, and under the additional assumption that $Z_t = Z_s + dZ_s$, where $\forall z_s \in Z_s$, $\|dz_s\| \leq r^*$, the function $g(\cdot)$ can be expressed using a Taylor series:

$$P_t = g(Z_t) = g(Z_s + dZ_s) = P_s + J_g(Z_s)dZ_s + o(dZ_s) \tag{24}$$

$$P_t - P_s = J_g(Z_s)dZ_s + o(dZ_s) \tag{25}$$

$$dZ_s = \frac{P_t - P_s - o(dZ_s)}{J_g(Z_s)} \tag{26}$$

$$\|dZ_s\|_F = \left\|\frac{P_t - P_s - o(dZ_s)}{J_g(Z_s)}\right\|_F \leq \left\|\frac{P_t - P_s}{J_g(Z_s)}\right\|_F + \left\|\frac{o(dZ_s)}{J_g(Z_s)}\right\|_F \leq \left\|\frac{P_t - P_s}{J_g(Z_s)}\right\|_F + \left\|\frac{o(kr^*)}{J_g(Z_s)}\right\|_F \tag{27}$$

Next, we examine the probability of the distance between $z_s$ and $z_t$ satisfying $\|dz_s\| \leq r^*$ under Assumption 3.2. Following the result from (Zhang et al., 2023), for any $x_t \in X_t$, and $r < r_t$, the probability distribution of $x_s$ falling within a ball $\mathcal{B}(x_t, r)$ of radius $r$ centered at $x_t$ is given by:

$$\mathbb{D}_s(x_s \in \mathcal{B}(x_t, r)) = \int_{\mathcal{B}(x_t,r)\cap\mathbb{D}_s} \frac{d\mathbb{D}_s}{d\lambda}(x_s)\, dx_s \geq \mu^- \lambda(\mathcal{B}(x_t, r) \cap \mathbb{D}_s) \geq c_t \mu^- \pi_{d_I} r^{d_I} \tag{28}$$

Let $\mathbb{I}(x_s \in \mathcal{B}(x_t, r))$ be an indicator function, where $\mathbb{I}(x_s \in \mathcal{B}(x_t, r))$ is independent and identically distributed Bernoulli random variables, representing the probability $\mathbb{D}_s(x_s \in \mathcal{B}(x_t, r))$. Let $S_n(x_t) = \sum_{i=1}^{n_s} \mathbb{I}(x_s \in B(x_t, r))$ denotes the number of source instances $x_s \in D_s$ that fall within $\mathcal{B}(x_t, r)$. Then, $S_n(x_t)$ follows a Binomial distribution. Let $W \sim \text{Binomial}(n_s, c_t\mu^-\pi_{d_I} r^{d_I})$. By applying Chernoff's inequality, we obtain the probability that the number of source data points falling within $\mathcal{B}(x_t, r)$ is less than $m$:

$$P(S_n(x_t) < m) = P(W < m) \leq \exp\left(-\frac{(E[W] - m)^2}{2E[W]}\right) = \exp\left(-\frac{(c_t\mu^-\pi_{d_I} r^{d_I} n_s - m)^2}{2c_t\mu^-\pi_{d_I} r^{d_I} n_s}\right) \tag{29}$$

Let $x_s^{(i)}$ denote the $i$-th nearest data point to $x_t$ within $B(x_t, r)$. The probability that the distance between $x_s^{(i)}$ and $x_t$ is less than $r$ is given by:

$$P(\|x_s^{(m)} - x_t\| \leq r) = P(S_n(x_t) \geq m) \geq 1 - \exp\left(-\frac{(c_t\mu^-\pi_{d_I} r^{d_I} n_s - m)^2}{2c_t\mu^-\pi_{d_I} r^{d_I} n_s}\right) \tag{30}$$

For a fixed $x_t$, it suffices to find a single nearest neighbor $x_s$ that lies within the ball $B(x_t, r)$, and thus we set $m = 1$. By applying the union bound, the desired probability can be expressed as follows:

$$
\begin{aligned}
&\bigcap_{x_t \in X_t} P(\|x_s^{(1)} - x_t\| \leq r) \\
&= \bigcap_{x_t \in X_t} P(S_n(x_t) \geq 1) \\
&= 1 - \bigcup_{x_t \in X_t} P(S_n(x_t) < 1) \\
&\geq 1 - k \exp\left(-\frac{(c_t \mu^- \pi_{d_I} r^{d_I} n_s - 1)^2}{2 c_t \mu^- \pi_{d_I} r^{d_I} n_s}\right) \\
&= 1 - \exp\left(-\frac{(c_t \mu^- \pi_{d_I} r^{d_I} n_s - 1)^2}{2 c_t \mu^- \pi_{d_I} r^{d_I} n_s} + \log k\right)
\end{aligned} \tag{31}
$$

Thus, with at least the probability $1 - \exp\left(-\frac{(c_t \mu^- \pi_{d_I} r^{d_I} n_s - 1)^2}{2 c_t \mu^- \pi_{d_I} r^{d_I} n_s} + \log k\right)$, the distance satisfies $\|dx_s\| \leq r \leq r_t$.

Finally, under Assumption 3.3, let $r = \frac{r^*}{L}$, then:

$$
\|dz_s\|_F \leq L \|dx_s\|_F \leq r^* \tag{32}
$$

Combining the above equations, with at least the probability:

$$
1 - \exp\left(-\frac{(c_t \mu^- \pi_{d_I} r^{d_I} n_s - 1)^2}{2 c_t \mu^- \pi_{d_I} r^{d_I} n_s} + \log k\right)
$$

we have:

$$
\|dZ_s\|_F \leq \left\| \frac{P_t - P_s}{J_g(Z_s)} \right\|_F + \left\| \frac{o(kr^*)}{J_g(Z_s)} \right\|_F \tag{33}
$$

This completes the proof of Equation (22).

Next, we prove Equation (23). Let $\Sigma_s'$ denote the correlation matrix computed from $k$ sampled source instances $Z_s$, and let $\hat{\Sigma}_s$ denote the pseudo-source correlation matrix computed from $k$ sampled test instances $Z_t$. These matrices are computed as follows:

$$
\Sigma_s' = Z_s^T Z_s \tag{34}
$$

$$
\hat{\Sigma}_s = Z_t^T Z_t = (Z_s + dZ_s)^T (Z_s + dZ_s) = Z_s^T Z_s + Z_s^T dZ_s + (dZ_s)^T Z_s + (dZ_s)^T dZ_s \tag{35}
$$

The change in the correlatione matrix is:

$$
\hat{\Sigma}_s - \Sigma_s' = Z_s^T dZ_s + (dZ_s)^T Z_s + (dZ_s)^T dZ_s \tag{36}
$$

Using the Frobenius norm, we obtain:

$$
\|\hat{\Sigma}_s - \Sigma_s'\|_F \leq \|Z_s^T dZ_s + (dZ_s)^T Z_s + (dZ_s)^T dZ_s\|_F \leq 2\|Z_s\|_F \|dZ_s\|_F + \|dZ_s\|_F^2 \tag{37}
$$

Additionally, since $\Sigma_s'$ is obtained from $k$ source domain instances and contains statistical error relative to the true covariance matrix $\Sigma_s = E[\Sigma_s']$. By Hoeffding's inequality, we have:

$$
P(\|\Sigma_s' - E[\Sigma_s']\|_F^2 \geq \epsilon) \leq 2 \exp\left(-\frac{2k\epsilon}{d^2}\right) \tag{38}
$$

Here, $d$ denotes the range of $\Sigma_s'$, which is set to 1. Let $2\exp\left(-\frac{2k\epsilon}{d^2}\right) = \sigma$, then:

$$\epsilon = -\frac{\log(\frac{\sigma}{2})}{2k} \tag{39}$$

With a probability of at least $1-\sigma$, we have:

$$\|\Sigma_s' - \Sigma_s\|_F < \sqrt{\epsilon} = \sqrt{\frac{\log(2/\delta)}{2k}} \tag{40}$$

By combining Equations (37) and (40), we obtain:

$$\|\Sigma_s - \hat{\Sigma}_s\|_F \leq \|\Sigma_s - \Sigma_s'\|_F + \|\Sigma_s' - \hat{\Sigma}_s\|_F \leq \sqrt{\frac{\log(2/\delta)}{2k}} + 2\|Z_s\|_F\|dZ_s\|_F + \|dZ_s\|_F^2 \tag{41}$$

We can further expand Equation (41) by applying Equation (33). However, since we cannot determine the true $P_s$ in Equation (33), we scale $\|P_t - P_s\|_F$ as follows:

$$\begin{aligned}
\|P_t - P_s\|_F &= \|P_t - \hat{Y}_t + \hat{Y}_t - l + l - P_s\|_F \\
&\leq \|P_t - \hat{Y}_t\|_F + \|\hat{Y}_t - l\|_F + \|l - P_s\|_F \\
&= \|P_t - \hat{Y}_t\|_F + \epsilon(h(X_t)) + \epsilon(h(X_s))
\end{aligned} \tag{42}$$

where $l$ is the true labels.

Finally, combining Equations (33), (41) and (42), we derive the following proposition: with at least $min(1 - \exp\left(-\frac{(c_t\mu^{-}\pi_{d_I} r^{d_I} n_s - 1)^2}{2c_t\mu^{-}\pi_{d_I} r^{d_I} n_s} + \log k\right), 1-\sigma)$:

$$\|\Sigma_s - \Sigma_t\|_F \leq 2\|Z_s\|_F\left(\frac{\|\hat{Y}_t - P_t\|_F + A}{\|J_g(Z_s)\|_F}\right) + \left(\frac{\|\hat{Y}_t - P_t\|_F + A}{\|J_g(Z_s)\|_F}\right)^2 + B \tag{43}$$

where $\hat{Y}_t$ is the one-hot encoding of $P_t$, $A = \|o(kr^*)\|_F + \epsilon(h(X_t)) + \epsilon(h(X_s))$ represents the output generalization error, and $B = \sqrt{\frac{\log(2/\delta)}{2k}}$ is the sampling error.

### B.2. Proof of Theorem 3.6

Here, we present Theorem 3.6 again for convenience.

**Theorem 3.6** Let $\mathcal{H}$ be a hypothesis class of VC-dimension $d_v$. If $\hat{h} \in \mathcal{H}$ minimizes the empirical error $\hat{\epsilon}_s(h)$ on $D_s$, and $h_t^* = \arg\min_{h\in\mathcal{H}} \epsilon_t(h)$ is the optimal hypothesis on $\mathbb{D}_t$, with the assumption that all hypotheses are L-Lipschitz continuous, then $\forall\delta \in (0,1)$, with probability with at least $1-\delta$ the following inequality holds:

$$\epsilon_t(\hat{h}) \leq \epsilon_t(h_t^*) + \mathcal{O}(\sqrt{\|\mu_s - \mu_t\|_F^2 + \|\Sigma_s - \Sigma_t\|_F^2}) + C$$

where $C = 2\sqrt{\frac{d_v log(2n_s) - \log(\delta)}{2n_s}} + 2\gamma$ and $\gamma = \min_{h\in\mathcal{H}}\{\epsilon_s(h(t)) + \epsilon_t(h(t))\}$. $\mu_s$, $\mu_t$, $\Sigma_s$ and $\Sigma_t$ denote the means and correlations of the source and test embeddings, respectively. We use $\mathcal{O}(\cdot)$ to hide the constant dependence.

To complete the proof, we begin by introducing some necessary definitions and assumptions.

**Definition B.1.** (Wasserstein Distance (Arjovsky et al., 2017)). The $\rho$-th order Wasserstein distance between two distributions $\mathbb{D}_s$ and $\mathbb{D}_t$ is defined as:

$$W_\rho(\mathbb{D}_s, \mathbb{D}_t) = \left( \inf_{\gamma \in \Pi[\mathbb{D}_s, \mathbb{D}_t]} \iint d(x_s, x_t)^\rho d\gamma(x_s, x_t) \right)^{1/\rho} \tag{44}$$

where $\Pi[\mathbb{D}_s, \mathbb{D}_t]$ is the set of all joint distributions on $\mathcal{X}_s \times \mathcal{X}_t$ with marginal distributions $\mathbb{D}_s$ and $\mathbb{D}_t$, and $d(x_s, x_u)$ is the distance function between two instances $x_s$ and $x_u$.

The Wasserstein distance can be intuitively understood in terms of the optimal transport problem, where $d(x_s, x_t)^\rho$ represents the unit cost of transporting mass from $x_s \in \mathbb{D}_s$ to $x_t \in \mathbb{D}_t$, and $\gamma(x_s, x_t)$ is the transport plan that satisfies the marginal constraints. According to the Kantorovich-Rubinstein theorem, the dual representation of the second-order Wasserstein distance can be written as:

$$\begin{aligned} &W_2(\mathbb{D}_s, \mathbb{D}_t) \\ &= \left( \inf_{\gamma \in \Pi[\mathbb{D}_s, \mathbb{D}_t]} \iint d(x_s, x_t)^2 d\gamma(x_s, x_t) \right)^{1/2} \\ &= \sup_{\|f\|_L \leq 1} (\|\mu_s - \mu_t\|_2^2 \\ &+ \text{tr}(\Sigma_s + \Sigma_t - 2(\Sigma_s^{1/2} \Sigma_t \Sigma_s^{1/2})^{1/2})^{1/2} \end{aligned} \tag{45}$$

where $\mu_s$ and $\mu_t$ are the means of $f(x_s)$ and $f(x_t)$, respectively, and $\|f\|_L = \sup \frac{|f(x_s) - f(x_t)|}{d(x_s, x_t)}$ is the Lipschitz semi-norm, which measures the rate of change of the function $f$ relative to the distance between $x_s$ and $x_t$. In this paper, we use $W_2$ as the default and omit the subscript 2. For completeness, we present Theorem 1 from (Shen et al., 2018) as follows:

**Lemma B.2.** *(Theorem 1 in (Shen et al., 2018)) Let $\mathcal{H}$ be an $L$-Lipschitz continuous hypothesis class with VC-dimension $d_v$. Given two domain distributions, $\mathbb{D}_s$ and $\mathbb{D}_t$, let $\gamma = \min_{h \in H}\{\epsilon_s(h(t)) + \epsilon_t(h(t))\}$. The risk of hypothesis $\hat{h}$ on the test domain is then bounded by:*

$$\epsilon_t(\hat{h}) \leq \gamma + \epsilon_s(\hat{h}) + 2LW(\mathbb{D}_s, \mathbb{D}_t) \tag{46}$$

From Definition B.1 and Lemma B.2, the difference between the true error on the training domain $\epsilon_s(h(t))$ and the true error on the test domain $\epsilon_t(h(t))$ can be obtained:

$$W(\mathbb{D}_S, \mathbb{D}_U) = \sqrt{\|\mu_s - \mu_t\|_2^2 + \text{tr}(\Sigma_s + \Sigma_t - 2(\Sigma_s^{1/2} \Sigma_t \Sigma_1^{1/2})^{1/2})} \leq \sqrt{\|\mu_s - \mu_t\|_F^2 + \|\Sigma_s - \Sigma_t\|_F^2} \tag{47}$$

$$|\epsilon_t(\hat{h}) - \epsilon_s(\hat{h})| \leq \gamma + 2L\sqrt{\|\mu_s - \mu_t\|_F^2 + \|\Sigma_s - \Sigma_t\|_F^2} \tag{48}$$

we use $\mathcal{O}$ to hide the constant dependence. Thus, we have:

$$|\epsilon_t(\hat{h}) - \epsilon_s(\hat{h})| \leq \gamma + \mathcal{O}(\sqrt{\|\mu_s - \mu_t\|_F^2 + \|\Sigma_s - \Sigma_t\|_F^2}) \tag{49}$$

Then, we provide an upper bound on the difference between the true error $\epsilon_s(h(t))$ and the empirical error $\hat{\epsilon}_s(h(t))$ on the source domain. We apply Lemma 7 of (Gui et al., 2024):

$$P[|\epsilon_t(\hat{h}) - \epsilon_s(\hat{h})| \geq \epsilon] \leq (2n_s)^{d_v} \exp(-2n_s\epsilon^2) \tag{50}$$

For any $\delta \in (0, 1)$, set $\delta = (2n_s)^{d_v} \exp(-2n_s\epsilon^2)$, we have:

$$\epsilon = \sqrt{\frac{d_v \log(2n_s) - \log \delta}{2n_s}} \tag{51}$$

Therefore, with probability at least $1-\delta$, we have:

$$|\hat{\epsilon}_s(\hat{h}) - \epsilon_s(\hat{h})| \leq \sqrt{\frac{d_v \log(2n_s) - \log\delta}{n_s}} \tag{52}$$

Combining Equations (49) and (52), let $h_j^*(t) = \arg\min_{h\in H} \epsilon_t(h)$, we obtain:

$$\begin{aligned}
&\epsilon_t(\hat{h}(t)) \\
&\leq \epsilon_s(\hat{h}(t)) + \gamma + \mathcal{O}\sqrt{\|\mu_s - \mu_t\|_2^2 + \|\Sigma_s - \Sigma_t\|_F^2} \\
&\leq \hat{\epsilon}_s(\hat{h}(t)) + \sqrt{\frac{d_v \log(2n_s) - \log\delta}{2n_s}} + \gamma + \mathcal{O}\sqrt{\|\mu_s - \mu_t\|_2^2 + \|\Sigma_s - \Sigma_t\|_F^2} \\
&\leq \hat{\epsilon}_s(h_t^*(t)) + \sqrt{\frac{d_v \log(2n_s) - \log\delta}{2n_s}} + \gamma + \mathcal{O}\sqrt{\|\mu_s - \mu_t\|_2^2 + \|\Sigma_s - \Sigma_t\|_F^2} \\
&\leq \epsilon_s(h_t^*(t)) + 2\sqrt{\frac{d_v \log(2n_s) - \log\delta}{2n_s}} + \gamma + \mathcal{O}\sqrt{\|\mu_s - \mu_t\|_2^2 + \|\Sigma_s - \Sigma_t\|_F^2} \\
&\leq \epsilon_t(h_t^*(t)) + 2\sqrt{\frac{d_v \log(2n_s) - \log\delta}{2n_s}} + 2\gamma + 2\mathcal{O}\sqrt{\|\mu_s - \mu_t\|_2^2 + \|\Sigma_s - \Sigma_t\|_F^2} \\
&= \epsilon_t(h_t^*(t)) + \mathcal{O}\sqrt{\|\mu_s - \mu_t\|_2^2 + \|\Sigma_s - \Sigma_t\|_F^2} + C
\end{aligned} \tag{53}$$

which completes the proof.

## C. Method Details

In this section, we describe the steps involved in the TCA algorithms used for test-time adaptation. The algorithm aligns feature correlations between the test and pseudo-source domains, without requiring access to the source domain data. The steps of the algorithm are outlined in Algorithm 1.

**Algorithm 1** LinearTCA Algorithm

1: **Input:** Test instances $X_t$, source model $h_\theta$.
2: **Output:** Final predictions $P_T^{'}$.
3: If use LinearTCA⁺: Update $\theta$ by Equation (1)
4: Obtain embeddings and predictions:

$$\hat{P}_t, Z_t = h_\theta(X_t)$$

5: Select $k$ high-certainty embeddings:

$$\hat{Z}_s = \{Z_t[i] \mid \omega_t^i \leq \omega_{min}^k\}$$

6: Compute linear transformation matrix $W$:

$$W = \text{argmin}_W \left\| W^T \Sigma_t W - \hat{\Sigma}_s \right\|_F^2$$

7: Apply transformation to embeddings:

$$Z_t^{'} = (Z_t - \mu_t)\, W + \hat{\mu}_s$$

8: Generate final predictions:

$$P_t^{'} = g(Z_t^{'})$$

## D. Experimental Details

### D.1. Datasets

The datasets used in this work consist of a variety of domain-shift challenges, enabling a comprehensive evaluation of test-time adaptation methods. The primary datasets employed include:

- **PACS**: The PACS dataset comprises 9,991 images across 7 distinct classes: {dog, elephant, giraffe, guitar, horse, house, person}. These images are drawn from four domains: {art, cartoons, photos, sketches }.
- **OfficeHome**: This dataset contains images from 4 different domains: {art, clipart, product, real-world}, with a total of 15,500 images. It includes 65 object categories, and the challenge lies in the significant domain shifts between the different visual styles. OfficeHome is widely used for evaluating domain generalization and adaptation methods due to its large number of categories and diverse image sources.
- **DomainNet**: The DomainNet dataset is a large-scale dataset used in transfer learning, consisting of 6 domains: {clipart, infograph, painting, quickdraw, real, and sketch}. It consists of a total of 586,575 images, with each domain containing 345 classes.
- **CIFAR-10/100C**: CIFAR-10 and CIFAR-100 are both foundational datasets in computer vision, containing 60,000 32x32 color images across 10 and 100 classes, respectively. The CIFAR-10/100C variants introduce additional corruptions (e.g., noise, blur, weather conditions) to simulate real-world distribution shifts, making them highly relevant for evaluating robustness under adversarial conditions.
- **ImageNet-C**: ImageNet-C is significantly larger compared to CIFAR10-C and CIFAR100-C. This dataset contains 1,281,167 training images and 50,000 test images, categorized into 1,000 classes. Like CIFAR10-C and CIFAR100-C, ImageNet-C also includes 15 types of corruptions.

### D.2. Backbones

The choice of backbone models is critical for the performance of domain adaptation algorithms, as they must efficiently extract features from images across various domains. For this work, we select the following backbone architectures:

- **ResNet-18/50**: ResNet-18 and ResNet-50 are used as backbone models in this study, where ResNet-18 offers a relatively lightweight model with fewer parameters, suitable for faster training and inference, while ResNet-50, with its deeper architecture, provides a more expressive feature representation that may improve performance on complex datasets.
- **ViT-B/16**: The Vision Transformer (ViT) is a more recent architecture that has demonstrated state-of-the-art performance in various vision tasks by treating images as sequences of patches. ViT-B/16 refers to a ViT model with a base configuration and a patch size of 16x16 pixels. ViT models are especially useful in scenarios where large-scale data and diverse domains are involved.
- **CLIP**: Contrastive Language- Image Pre-Training (CLIP), developed by OpenAI, is a cutting-edge multimodal model that bridges visual and textual domains through contrastive learning. CLIP employs dual encoders (ResNet/ViT for images and Transformer for text) to project both modalities into a shared semantic space, enabling zero-shot classification by matching image features with natural language prompts.

Both ResNet and ViT backbones are well-established in the literature and serve as strong candidates for evaluating domain adaptation techniques, with ResNet-18/50 being more computationally efficient and ViT-B/16 being particularly effective in capturing complex relationships across domains. In this work, the zero-shot classification model CLIP is also included as a backbone to validate the effectiveness of our proposed methods on closed-source foundation models.

### D.3. Implementation Details

**Consistent with prior work (Wang et al., 2020; Niu et al., 2022; 2023; Nguyen et al., 2023; Yuan et al., 2024; Iwasawa & Matsuo, 2021; Wang et al., 2023; Zhang et al., 2023), hyperparameter tuning in our experimental setup is conducted**

**across datasets.** Specifically, in the Domain Generalization task, we first identify the optimal parameter set based on the highest accuracy achieved on the default domain (art paintings in PACS, art in OfficeHome and clipart in DomainNet). These parameters are then applied to other domains to assess their performance. Specifically, we conduct a search for the learning rate within the range {1e-7, 5e-7, 1e-6, 5e-6, 1e-5, 5e-5, 1e-4, 5e-4, 1e-3, 5e-3, 1e-2, 5e-2, 1e-1}. For methods that include an entropy filter component (e.g., TSD), we explore the entropy filter hyperparameter in the set {1, 5, 10, 15, 20, 50, 100, 200, 300}. For AdaNPC, we explore the hyperparameter $k$ (the number of nearest samples used for voting) over {5, 10, 15, 20, 30, 40, 50}. For the LinearTCA method, we optimized the number of pseudo-source instances $k$ within the range {5, 10, 15, 20, 25, 30, 35, 40, 45, 50, 100, 200, 300}. For most datasets and backbones, smaller $k$ values generally yield satisfactory results. For datasets with a substantial number of images per class, it is advisable to experiment with larger $k$ values. For the LinearTCA$^+$ method, we conducted an optimization of $k$ values on the basis of other top-performing test-time adaptation method and its parameter settings.

For the Image Corruption task, we experiment with each TTA method using learning rates from {1e-7, 5e-7, 1e-6, 5e-6, 1e-5, 5e-5, 1e-4, 5e-4, 1e-3, 5e-3, 1e-2, 5e-2, 1e-1} and the entropy filter hyperparameter in the set {1, 5, 10, 15, 20, 50, 100, 200, 300}. The parameter range for $k$ in AdaNPC, LinearTCA/LinearTCA$^+$ remains consistent with their respective selections in Domain Generalization task. The top-performing test-time adaptation approach on the Image Corruption is selected as the base method for LinearTCA$^+$. The best performance results obtained for each method are selected as the final experimental outcomes. For the pre-trained model on ImageNet-C dataset, we utilize the model provided by TorchVision.

During the Test-Time Adaptation phase, both the Domain Generalization and Image Corruption tasks utilize specific batch size for different backbones. ResNet-18 and ResNet-50 use a batch size of 128, whereas the ViT-B/16 is configured with a batch size of 64.

For the implementation of the TCA method, we first obtain the embeddings of all test data during the testing phase. Based on the inter-class proportion of the test data, we perform high-certainty filtering to select instances that match this proportion to construct the pseudo-source domain. Subsequently, we use the correlation distance between the pseudo-source domain and the test domain to compute the linear transformation matrix $W$. Finally, we apply this linear transformation to the previously retained embeddings of the test data and make final prediction.

## E. Additional Experimental Results

### E.1. Comparison Results Details

Tables 9 to 17 provide the detailed results of our experimental results on Domain Generalization task, and Tables 18 to 26 offers a detailed overview of the outcomes from our Image Corruption task. These results demonstrate that our TCA method consistently outperforms other state-of-the-art TTA approaches across most domians and corruption types, effectively validating the TCA's capability to robustly enhance accuracy performance during the test phase.

### E.2. Analysis Details

Figures 5 and 6 illustrate the adaptation process of LinearTCA to datasets with linear and nonlinear shifts, respectively. Figures (a) to (f) depict the gradual alignment process of linear and nonlinear shifts. Notably, LinearTCA demonstrates significantly better performance in adapting to linear shifts compared to nonlinear ones, which the LinearTCA's proficiency in handling simpler, linear distribution shifts while revealing its limitations when addressing more complex, nonlinear transformations.

We also provide the code for generating source and target domain features with both linear and nonlinear distribution shifts. The features are generated using PyTorch and serve as synthetic examples. The source domain features ($X_s$, $X_s^{(2)}$) consist of clusters sampled from normal distributions with fixed offsets. The target domain features ($X_t$, $X_t^{(2)}$) are scaled and shifted versions of normal distributions to simulate linear and nonlinear domain shifts. The generated features can be visualized using 2D scatter plots for better understanding of the distributional changes.

**Linear Shift Code:**

```
# Linear Shift
  # Source domain features
X_s = torch.cat((torch.randn(30, 2),
                 torch.randn(30, 2) + 15,
                 torch.randn(30, 2) + torch.tensor([0, 10])), dim=0)
  # Target domain features
X_t = torch.cat((torch.randn(250, 2) * 2 + 7,
                 torch.randn(250, 2) * 2.5 + torch.tensor([0, 20]),
                 torch.randn(250, 2) * 3 + 21), dim=0)
```

**Nonlinear Shift Code:**

```
# Nonlinear Shift
  # Source domain features
X_s_2 = torch.cat((torch.randn(30, 2),
                   torch.randn(30, 2) + 10,
                   torch.randn(30, 2) + torch.tensor([0, 10]),
                   torch.randn(30, 2) + torch.tensor([-5, -10])), dim=0)
  # Target domain features
X_t_2 = torch.cat((torch.randn(250, 2) * 3 + 5,
                   torch.randn(250, 2) + 10,
                   torch.randn(250, 2) * 2 + torch.tensor([0, 20]),
                   torch.randn(250, 2) * 2.5 + torch.tensor([-9, 1])), dim=0)
```

| Backbone | Method | PACS | | | | Avg | Hyper-parameters |
|---|---|---|---|---|---|---|---|
| | | A | C | P | S | | |
| ResNet-18 | Source (He et al., 2016) | 78.37 | 77.39 | 95.03 | 76.58 | 81.84 | nan |
| | BN (Schneider et al., 2020) | 80.91 | 80.80 | 95.09 | 73.81 | 82.65 | nan |
| | T3A (Iwasawa & Matsuo, 2021) | 80.27 | 79.56 | 95.57 | 77.40 | 83.20 | fk=50 |
| | AdaNPC (Zhang et al., 2023) | 80.81 | 79.14 | 96.17 | 77.81 | 83.48 | fk=100 k=5 |
| | TENT (Wang et al., 2020) | 82.86 | 82.12 | 96.11 | 79.82 | 85.23 | lr=5e-3 |
| | PLC (Lee, 2013) | 81.69 | 81.36 | 95.87 | 73.71 | 83.16 | lr=1e-3 |
| | EATA (Niu et al., 2022) | 82.71 | 81.36 | 94.79 | 74.34 | 83.30 | lr=1e-2 |
| | SAR (Niu et al., 2023) | 83.30 | 82.55 | 95.09 | 80.68 | 85.41 | lr=1e-1 |
| | TIPI (Nguyen et al., 2023) | 85.50 | 84.90 | 96.05 | **83.13** | 87.39 | lr=5e-3 |
| | TEA (Yuan et al., 2024) | 86.47 | 85.79 | 95.69 | 80.81 | 87.19 | lr=5e-3 |
| | TSD (Wang et al., 2023) | 86.96 | 86.73 | 96.41 | 81.22 | 87.83 | lr=1e-4 fk=100 |
| | LinearTCA | 80.91 | 81.02 | 95.69 | 76.74 | 83.59 | fkTCA=30 |
| | LinearTCA $^{+}$ | **88.38** | **87.12** | **96.59** | 83.00 | **88.77** | TSD fkTCA=25 |

*Table 9.* Accuracy comparison of different TTA methods on PACS dataset based on ResNet-18 backbone. The best results are highlighted in **boldface**, and the second ones are underlined.

| Backbone | Method | PACS | | | | Avg | Hyper-parameters |
|---|---|---|---|---|---|---|---|
| | | A | C | P | S | | |
| ResNet-50 | Source (He et al., 2016) | 83.89 | 81.02 | 96.17 | 78.04 | 84.78 | nan |
| | BN (Schneider et al., 2020) | 85.50 | 85.62 | 96.77 | 72.05 | 84.99 | nan |
| | T3A (Iwasawa & Matsuo, 2021) | 84.86 | 82.47 | 97.01 | 78.52 | 85.71 | fk=100 |
| | AdaNPC (Zhang et al., 2023) | 85.11 | 82.85 | 97.13 | 79.41 | 86.12 | fk=200 k=10 |
| | TENT (Wang et al., 2020) | 88.09 | 87.33 | 97.19 | 79.69 | 88.07 | lr=1e-3 |
| | PLC (Lee, 2013) | 86.52 | 84.94 | 97.01 | 77.88 | 86.59 | lr=1e-3 |
| | EATA (Niu et al., 2022) | 84.72 | 85.20 | 96.35 | 72.46 | 84.68 | lr=5e-5 |
| | SAR (Niu et al., 2023) | 85.55 | 85.62 | 96.77 | 75.24 | 85.79 | lr=1e-2 |
| | TIPI (Nguyen et al., 2023) | 88.18 | 87.93 | 97.13 | 78.80 | 88.01 | lr=1e-3 |
| | TEA (Yuan et al., 2024) | 88.67 | 87.80 | 97.54 | 80.99 | 88.75 | lr=1e-3 |
| | TSD (Wang et al., 2023) | 90.43 | 89.89 | **97.84** | 81.80 | 89.99 | lr=1e-4 fk=100 |
| | LinearTCA | 86.28 | 83.92 | 96.95 | 79.99 | 86.78 | fkTCA=30 |
| | LinearTCA + | **90.92** | **90.10** | **97.84** | **83.86** | **90.68** | TSD fkTCA=30 |

*Table 10.* Accuracy comparison of different TTA methods on PACS dataset based on ResNet-50 backbone. The best results are highlighted in **boldface**, and the second ones are underlined.

| Backbone | Method | PACS | | | | Avg | Hyper-parameters |
|---|---|---|---|---|---|---|---|
| | | A | C | P | S | | |
| ViT-B/16 | Source (He et al., 2016) | 86.96 | 84.30 | 98.02 | 78.77 | 87.02 | nan |
| | BN (Schneider et al., 2020) | 0.00 | 0.00 | 0.00 | 0.00 | 0.00 | nan |
| | T3A (Iwasawa & Matsuo, 2021) | 88.23 | 85.96 | 98.86 | 79.18 | 88.06 | fk=50 |
| | AdaNPC (Zhang et al., 2023) | 89.01 | **87.37** | **98.98** | 81.06 | 89.11 | fk=200 k=10 |
| | TENT (Wang et al., 2020) | 89.60 | 73.08 | 97.90 | 79.33 | 84.98 | lr=5e-3 |
| | PLC (Lee, 2013) | 87.70 | 85.28 | 98.62 | 80.30 | 87.97 | lr=5e-4 |
| | EATA (Niu et al., 2022) | 87.45 | 84.17 | 97.84 | 76.92 | 86.60 | lr=5e-3 |
| | SAR (Niu et al., 2023) | 86.96 | 84.30 | 98.02 | 79.18 | 87.12 | lr=5e-2 |
| | TIPI (Nguyen et al., 2023) | 87.99 | 84.17 | 98.20 | 81.55 | 87.98 | lr=5e-4 |
| | TEA (Yuan et al., 2024) | 88.77 | 85.41 | 97.96 | 77.35 | 87.37 | lr=1e-3 |
| | TSD (Wang et al., 2023) | **90.72** | 85.41 | 97.96 | 59.63 | 83.43 | lr=1e-5 fk=20 |
| | LinearTCA | 88.57 | 86.52 | 98.26 | 81.09 | 88.61 | fkTCA=15 |
| | LinearTCA + | 88.96 | 86.90 | 98.26 | **83.05** | **89.30** | TIPI fkTCA=30 |

*Table 11.* Accuracy comparison of different TTA methods on PACS dataset based on ViT-B/16 backbone. The best results are highlighted in **boldface**, and the second ones are underlined.

| Backbone | Method | OfficeHome | | | | Avg | Hyper-parameters |
|---|---|---|---|---|---|---|---|
| | | A | C | P | R | | |
| ResNet-18 | Source (He et al., 2016) | 56.45 | 48.02 | 71.34 | 72.23 | 62.01 | nan |
| | BN (Schneider et al., 2020) | 55.62 | 49.32 | 70.60 | 72.66 | 62.05 | nan |
| | T3A (Iwasawa & Matsuo, 2021) | 56.61 | 50.06 | **73.39** | 72.99 | 63.26 | fk=20 |
| | AdaNPC (Zhang et al., 2023) | 55.95 | 49.42 | 73.10 | 73.05 | 62.88 | fk=20 k=5 |
| | TENT (Wang et al., 2020) | 56.94 | 50.65 | 71.86 | 72.92 | 63.09 | lr=1e-3 |
| | PLC (Lee, 2013) | 55.95 | 49.37 | 70.83 | 72.73 | 62.22 | lr=5e-5 |
| | EATA (Niu et al., 2022) | 56.41 | 49.62 | 71.66 | 72.27 | 62.49 | lr=1e-3 |
| | SAR (Niu et al., 2023) | 57.15 | 50.31 | 70.24 | 72.34 | 62.51 | lr=5e-2 |
| | TIPI (Nguyen et al., 2023) | 57.03 | 50.61 | 72.07 | **73.28** | 63.25 | lr=1e-3 |
| | TEA (Yuan et al., 2024) | 58.55 | 50.47 | 71.75 | 72.94 | 63.43 | lr=5e-4 |
| | TSD (Wang et al., 2023) | 58.06 | 49.81 | 71.37 | 70.67 | 62.47 | lr=1e-4 fk=10 |
| | LinearTCA | 59.46 | 50.40 | 72.02 | 72.78 | 63.66 | fkTCA=10 |
| | LinearTCA + | **59.83** | **51.80** | 72.29 | 73.17 | **64.27** | TEA fkTCA=10 |

*Table 12.* Accuracy comparison of different TTA methods on OfficeHome dataset based on ResNet-18 backbone. The best results are highlighted in **boldface**, and the second ones are underlined.

| Backbone | Method | OfficeHome | | | | Avg | Hyper-parameters |
|---|---|---|---|---|---|---|---|
| | | A | C | P | R | | |
| ResNet-50 | Source (He et al., 2016) | 64.85 | 52.26 | 75.04 | 75.88 | 67.01 | nan |
| | BN (Schneider et al., 2020) | 63.54 | 52.71 | 73.89 | 75.05 | 66.30 | nan |
| | T3A (Iwasawa & Matsuo, 2021) | 65.02 | 53.31 | 76.10 | 76.96 | 67.85 | fk=100 |
| | AdaNPC (Zhang et al., 2023) | 63.74 | 52.33 | 75.72 | 76.43 | 67.05 | fk=200 k=5 |
| | TENT (Wang et al., 2020) | 64.65 | 54.85 | 75.04 | 76.15 | 67.67 | lr=5e-4 |
| | PLC (Lee, 2013) | 63.82 | 52.83 | 74.09 | 75.03 | 66.44 | lr=5e-5 |
| | EATA (Niu et al., 2022) | 63.95 | 53.95 | 74.57 | 75.56 | 67.01 | lr=1e-3 |
| | SAR (Niu et al., 2023) | 64.77 | 55.92 | 75.24 | 75.81 | 67.94 | lr=1e-2 |
| | TIPI (Nguyen et al., 2023) | 64.73 | 56.24 | 75.47 | 77.00 | 68.36 | lr=1e-3 |
| | TEA (Yuan et al., 2024) | 65.97 | **57.57** | 74.72 | 75.97 | 68.56 | lr=1e-3 |
| | TSD (Wang et al., 2023) | 65.51 | 56.54 | 76.17 | 76.31 | 68.63 | lr=1e-4 fk=1 |
| | LinearTCA | 66.50 | 54.39 | 75.76 | **77.07** | 68.43 | fkTCA=5 |
| | LinearTCA [+] | **67.16** | 56.22 | **76.86** | 77.05 | **69.32** | TSD fkTCA=10 |

*Table 13.* Accuracy comparison of different TTA methods on OfficeHome dataset based on ResNet-50 backbone. The best results are highlighted in **boldface**, and the second ones are underlined.

| Backbone | Method | OfficeHome | | | | Avg | Hyper-parameters |
|---|---|---|---|---|---|---|---|
| | | A | C | P | R | | |
| ViT-B/16 | Source (He et al., 2016) | 73.51 | 63.18 | 82.68 | 85.06 | 76.11 | nan |
| | BN (Schneider et al., 2020) | 0.00 | 0.00 | 0.00 | 0.00 | 0.00 | nan |
| | T3A (Iwasawa & Matsuo, 2021) | **77.79** | 65.57 | **85.92** | **86.18** | 78.87 | fk=5 |
| | AdaNPC (Zhang et al., 2023) | 75.57 | 63.76 | 84.30 | 85.43 | 77.26 | fk=200 k=5 |
| | TENT (Wang et al., 2020) | 74.58 | 64.15 | 83.74 | 85.36 | 76.95 | lr=1e-3 |
| | PLC (Lee, 2013) | 74.41 | 63.51 | 82.81 | 85.31 | 76.51 | lr=1e-4 |
| | EATA (Niu et al., 2022) | 74.17 | 64.81 | 83.58 | 85.38 | 76.98 | lr=1e-3 |
| | SAR (Niu et al., 2023) | 74.95 | 63.07 | 83.58 | 85.06 | 76.66 | lr=1e-1 |
| | TIPI (Nguyen et al., 2023) | 74.50 | 64.47 | 83.92 | 85.49 | 77.09 | lr=1e-3 |
| | TEA (Yuan et al., 2024) | 73.71 | 63.23 | 82.74 | 84.92 | 76.15 | lr=1e-4 |
| | TSD (Wang et al., 2023) | 75.94 | 55.95 | 84.75 | 85.33 | 75.49 | lr=1e-5 fk=20 |
| | LinearTCA | 76.02 | 67.35 | 84.12 | 85.56 | 78.26 | fkTCA=5 |
| | LinearTCA [+] | 77.21 | **68.36** | 84.64 | 85.88 | **79.02** | TIPI fkTCA=5 |

*Table 14.* Accuracy comparison of different TTA methods on OfficeHome dataset based on ViT-B/16 backbone. The best results are highlighted in **boldface**, and the second ones are underlined.

| Backbone | Method | DomainNet | | | | | | Avg | Hyper-parameters |
|---|---|---|---|---|---|---|---|---|---|
| | | C | I | P | Q | R | S | | |
| ResNet-18 | Source (He et al., 2016) | 57.30 | 16.86 | 45.03 | 12.69 | 56.89 | 46.00 | 39.13 | nan |
| | BN (Schneider et al., 2020) | 57.26 | 11.55 | 43.32 | 11.77 | 56.58 | 47.09 | 37.93 | nan |
| | T3A (Iwasawa & Matsuo, 2021) | 58.44 | 18.57 | 46.80 | 14.54 | 57.66 | 47.72 | 40.62 | fk=100 |
| | AdaNPC (Zhang et al., 2023) | 57.61 | 15.83 | 44.89 | **18.44** | **59.72** | 46.53 | 40.50 | fk=100 k=10 |
| | TENT (Wang et al., 2020) | 58.41 | 13.09 | 45.17 | 13.02 | 57.89 | 48.94 | 39.42 | lr=1e-4 |
| | PLC (Lee, 2013) | 57.45 | 12.60 | 44.77 | 10.01 | 55.74 | 47.20 | 37.96 | lr=1e-5 |
| | EATA (Niu et al., 2022) | 59.18 | 16.22 | 46.65 | 18.04 | 59.59 | 50.21 | 41.65 | lr=1e-3 |
| | SAR (Niu et al., 2023) | 59.13 | 13.10 | 45.75 | 4.88 | 58.25 | 49.83 | 38.49 | lr=5e-3 |
| | TIPI (Nguyen et al., 2023) | 58.42 | 11.68 | 42.53 | 5.37 | 50.76 | 47.58 | 36.05 | lr=5e-4 |
| | TEA (Yuan et al., 2024) | 58.01 | 12.83 | 45.10 | 14.33 | 57.55 | 48.80 | 39.43 | lr=5e-5 |
| | TSD (Wang et al., 2023) | 57.73 | 12.19 | 44.58 | 12.78 | 55.94 | 48.31 | 38.59 | lr=1e-5 fk=100 |
| | LinearTCA | 58.67 | **18.60** | 46.85 | 14.88 | 57.93 | 47.80 | 40.79 | fkTCA=10 |
| | LinearTCA [+] | **59.95** | 16.89 | **47.68** | 18.35 | 59.67 | **50.66** | **42.20** | EATA fkTCA=10 |

*Table 15.* Accuracy comparison of different TTA methods on DomainNet dataset based on ResNet-18 backbone. The best results are highlighted in **boldface**, and the second ones are underlined.

| Backbone | Method | DomainNet | | | | | | Avg | Hyper-parameters |
|---|---|---|---|---|---|---|---|---|---|
| | | C | I | P | Q | R | S | | |
| ResNet-50 | Source (He et al., 2016) | 63.68 | 20.93 | 50.35 | 12.95 | 62.16 | 51.42 | 43.58 | nan |
| | BN (Schneider et al., 2020) | 63.30 | 14.84 | 48.54 | 10.83 | 62.02 | 52.12 | 41.94 | nan |
| | T3A (Iwasawa & Matsuo, 2021) | 63.76 | 21.06 | 49.82 | 18.46 | 64.05 | 52.39 | 44.92 | fk=100 |
| | AdaNPC (Zhang et al., 2023) | 64.38 | 20.12 | 51.07 | 17.34 | 65.59 | 52.51 | 45.17 | fk=200 k=10 |
| | TENT (Wang et al., 2020) | 64.95 | 17.46 | 51.58 | 11.28 | 64.04 | 54.51 | 43.97 | lr=1e-4 |
| | PLC (Lee, 2013) | 63.56 | 14.89 | 49.09 | 8.83 | 60.83 | 52.61 | 41.63 | lr=1e-5 |
| | EATA (Niu et al., 2022) | 65.89 | 19.88 | **52.67** | 20.36 | 66.58 | 55.99 | 46.89 | lr=5e-4 |
| | SAR (Niu et al., 2023) | 65.44 | 14.63 | 50.68 | 3.94 | 63.94 | 54.49 | 42.19 | lr=5e-3 |
| | TIPI (Nguyen et al., 2023) | 64.97 | 17.47 | 51.63 | 11.67 | 64.03 | 54.69 | 44.08 | lr=1e-4 |
| | TEA (Yuan et al., 2024) | 64.87 | 16.95 | 51.40 | 11.48 | 61.90 | 54.31 | 43.48 | lr=1e-4 |
| | TSD (Wang et al., 2023) | 64.31 | 16.53 | 50.75 | 8.52 | 58.97 | 53.63 | 42.12 | lr=5e-5 fk=5 |
| | LinearTCA | 64.58 | **23.79** | 50.06 | 14.10 | 63.60 | 53.21 | 44.89 | fkTCA=5 |
| | LinearTCA + | **66.46** | 21.04 | 51.61 | **20.47** | **66.86** | **56.57** | **47.17** | EATA fkTCA=5 |

*Table 16.* Accuracy comparison of different TTA methods on DomainNet dataset based on ResNet-50 backbone. The best results are highlighted in **boldface**, and the second ones are underlined.

| Backbone | Method | DomainNet | | | | | | Avg | Hyper-parameters |
|---|---|---|---|---|---|---|---|---|---|
| | | C | I | P | Q | R | S | | |
| ViT-B/16 | Source (He et al., 2016) | 71.62 | 25.59 | 57.34 | 18.07 | 71.90 | 57.24 | 50.29 | nan |
| | BN (Schneider et al., 2020) | 0.00 | 0.00 | 0.00 | 0.00 | 0.00 | 0.00 | 0.00 | nan |
| | T3A (Iwasawa & Matsuo, 2021) | 73.56 | 26.95 | 59.77 | **27.34** | 75.77 | 60.24 | 53.94 | fk=100 |
| | AdaNPC (Zhang et al., 2023) | 73.76 | 25.52 | 59.86 | 24.42 | 75.99 | 60.14 | 53.28 | fk=200 k=5 |
| | TENT (Wang et al., 2020) | 72.65 | 18.00 | 35.08 | 4.20 | 74.03 | 35.82 | 39.96 | lr=1e-4 |
| | PLC (Lee, 2013) | 72.29 | 19.27 | 56.69 | 5.00 | 72.45 | 58.04 | 47.29 | lr=5e-5 |
| | EATA (Niu et al., 2022) | 73.91 | 28.76 | 61.71 | 24.79 | 75.39 | 61.84 | 54.40 | lr=1e-3 |
| | SAR (Niu et al., 2023) | 73.06 | 17.42 | 40.94 | 11.37 | 73.67 | 40.39 | 42.81 | lr=5e-2 |
| | TIPI (Nguyen et al., 2023) | 72.71 | 17.62 | 33.37 | 4.54 | 73.98 | 35.96 | 39.70 | lr=1e-4 |
| | TEA (Yuan et al., 2024) | 71.96 | 24.17 | 55.31 | 8.83 | 72.20 | 58.00 | 48.41 | lr=5e-5 |
| | TSD (Wang et al., 2023) | 72.40 | 23.47 | 59.20 | 4.12 | 73.54 | 59.60 | 48.72 | lr=1e-6 fk=50 |
| | LinearTCA | 73.37 | 28.42 | 60.88 | 20.66 | 73.39 | 60.03 | 52.79 | fkTCA=5 |
| | LinearTCA + | **75.02** | **30.11** | **63.33** | 25.14 | **76.05** | **63.31** | **55.49** | EATA fkTCA=5 |

*Table 17.* Accuracy comparison of different TTA methods on DomainNet dataset based on ViT-B/16 backbone. The best results are highlighted in **boldface**, and the second ones are underlined.

| Method | CIFAR-10-C | | | | | | | | | | | | | | | Avg |
|---|---|---|---|---|---|---|---|---|---|---|---|---|---|---|---|---|
| | Gau. | Sho. | Imp. | Def. | Gla. | Mot. | Zoo. | Sno. | Fro. | Fog | Bri. | Con. | Ela. | Pix. | Jpe. | |
| Source (He et al., 2016) | 27.43 | 33.56 | 21.57 | 43.64 | 40.48 | 51.26 | 51.29 | 68.18 | 54.52 | 66.65 | 87.50 | 27.59 | 67.06 | 48.86 | 72.37 | 50.80 |
| BN (Schneider et al., 2020) | 66.05 | 68.22 | 56.83 | 82.34 | 57.86 | 79.78 | 82.32 | 74.99 | 74.30 | 78.85 | 87.22 | 81.80 | 70.31 | 73.61 | 71.00 | 73.70 |
| T3A (Iwasawa & Matsuo, 2021) | 44.16 | 50.32 | 29.64 | 56.98 | 49.02 | 60.85 | 62.29 | 70.20 | 60.83 | 70.75 | 87.23 | 37.68 | 71.60 | 58.59 | 73.22 | 58.89 |
| AdaNPC (Zhang et al., 2023) | 40.96 | 47.54 | 27.01 | 54.84 | 46.47 | 59.93 | 61.91 | 70.12 | 60.59 | 71.18 | 87.16 | 35.13 | 71.06 | 58.15 | 73.73 | 57.72 |
| TENT (Wang et al., 2020) | 65.09 | 72.78 | 58.93 | 82.78 | 59.02 | 81.01 | 83.92 | 77.82 | 75.83 | 79.34 | 88.10 | 82.77 | 72.10 | 76.47 | 72.26 | 75.21 |
| PLC (Lee, 2013) | 66.06 | 68.25 | 56.92 | 82.66 | 57.69 | 79.78 | 82.29 | 74.84 | 74.33 | 78.91 | 87.07 | 81.82 | 70.49 | 73.63 | 71.00 | 73.72 |
| EATA (Niu et al., 2022) | 66.89 | 68.21 | 56.76 | 82.49 | 57.59 | 80.10 | 82.09 | 74.90 | 74.35 | 78.82 | 87.13 | 82.04 | 70.66 | 74.16 | 71.73 | 73.86 |
| SAR (Niu et al., 2023) | 66.28 | 68.23 | 58.30 | 82.34 | 59.20 | 79.78 | 82.32 | 74.99 | 74.53 | 78.85 | 87.22 | 82.51 | 70.32 | 73.61 | 71.00 | 73.97 |
| TIPI (Nguyen et al., 2023) | 67.69 | 73.21 | 59.54 | **83.80** | **62.36** | 81.29 | **84.15** | 78.15 | **76.90** | 79.91 | **88.63** | **82.99** | 72.46 | 77.34 | 73.11 | 76.10 |
| TEA (Yuan et al., 2024) | 70.76 | 72.46 | 61.44 | 83.40 | 60.45 | 81.56 | 84.05 | 77.57 | 76.12 | 81.07 | 87.97 | 82.82 | 72.51 | 76.51 | 74.26 | 76.20 |
| TSD (Wang et al., 2023) | 72.33 | 75.73 | 64.84 | 83.24 | 61.45 | **82.49** | 83.92 | 78.29 | 75.79 | 81.96 | 87.55 | 79.43 | 73.07 | 78.48 | 75.36 | 76.93 |
| LinearTCA | 52.17 | 55.61 | 36.34 | 57.08 | 48.18 | 62.25 | 62.26 | 71.94 | 67.17 | 73.09 | 87.23 | 41.70 | 70.28 | 56.43 | 72.68 | 60.96 |
| LinearTCA + | **73.11** | **75.93** | **65.30** | 83.23 | 62.13 | 82.21 | 83.87 | **78.41** | 76.25 | **82.12** | 87.42 | 79.32 | **73.48** | **78.60** | **75.62** | **77.13** |

*Table 18.* Accuracy comparisons of different TTA methods on CIFAR-10-C dataset at damage level of 5, based on ResNet-18 backbone. The best results are highlighted in **boldface**, and the second ones are underlined.

| Method | CIFAR-10-C | | | | | | | | | | | | | | | Avg |
|---|---|---|---|---|---|---|---|---|---|---|---|---|---|---|---|---|
| | Gau. | Sho. | Imp. | Def. | Gla. | Mot. | Zoo. | Sno. | Fro. | Fog | Bri. | Con. | Ela. | Pix. | Jpe. | |
| Source (He et al., 2016) | 30.81 | 37.09 | 24.71 | 38.07 | 41.66 | 51.97 | 51.17 | 68.49 | 60.52 | 66.79 | 86.19 | 28.25 | 65.19 | 38.95 | 71.66 | 50.77 |
| BN (Schneider et al., 2020) | 61.98 | 63.05 | 56.25 | 82.58 | 54.49 | 80.11 | 82.61 | 74.16 | 72.36 | 79.28 | 87.04 | 81.06 | 67.16 | 71.27 | 70.22 | 72.24 |
| T3A (Iwasawa & Matsuo, 2021) | 45.34 | 49.51 | 36.76 | 39.10 | 46.88 | 56.85 | 53.52 | 65.88 | 57.84 | 68.82 | 84.61 | 33.13 | 68.20 | 46.08 | 70.49 | 54.87 |
| AdaNPC (Zhang et al., 2023) | 41.93 | 47.29 | 32.50 | 41.60 | 45.70 | 56.09 | 56.38 | 67.25 | 59.62 | 69.90 | 85.51 | 32.24 | 68.27 | 46.08 | 70.88 | 54.75 |
| TENT (Wang et al., 2020) | 62.04 | 63.30 | 56.26 | 82.66 | 54.52 | 80.09 | 82.68 | 74.40 | 72.43 | 79.20 | 87.21 | 81.11 | 67.34 | 71.39 | 70.32 | 72.33 |
| PLC (Lee, 2013) | 62.35 | 62.71 | 56.09 | 82.57 | 54.07 | 80.24 | 82.91 | 74.54 | 72.26 | 79.37 | 87.20 | 81.09 | 67.62 | 71.39 | 70.71 | 72.34 |
| EATA (Niu et al., 2022) | 62.61 | 63.63 | 56.13 | 82.34 | 54.71 | 79.97 | 82.16 | 74.89 | 72.16 | 79.27 | 87.66 | 81.32 | 67.76 | 70.81 | 70.28 | 72.38 |
| SAR (Niu et al., 2023) | 65.12 | 66.49 | 58.49 | 82.58 | 55.65 | 80.12 | 82.61 | 75.10 | **73.60** | 79.63 | 87.04 | **81.56** | 68.49 | 72.63 | 71.47 | 73.37 |
| TIPI (Nguyen et al., 2023) | 62.02 | 63.61 | 55.37 | 82.80 | 54.43 | 80.29 | 83.11 | 74.81 | 72.77 | 78.96 | 87.52 | 81.35 | 67.49 | 71.72 | 70.70 | 72.46 |
| TEA (Yuan et al., 2024) | 63.92 | 65.15 | 55.73 | 82.32 | 52.34 | 80.54 | 83.14 | 74.99 | 73.17 | 80.08 | 87.58 | 80.90 | 67.57 | 70.47 | 70.26 | 72.54 |
| TSD (Wang et al., 2023) | 64.42 | 65.56 | 56.16 | **83.06** | 53.95 | **80.88** | **83.32** | **75.18** | 73.58 | **80.17** | **87.84** | 81.49 | 68.38 | **72.91** | 71.61 | 73.23 |
| LinearTCA | 52.05 | 55.76 | 43.06 | 51.79 | 49.06 | 61.68 | 62.03 | 71.53 | 67.67 | 72.83 | 86.04 | 37.62 | **69.92** | 50.28 | **72.69** | 60.27 |
| LinearTCA + | **65.27** | **66.63** | **59.15** | 82.87 | **56.37** | 80.78 | 82.80 | 75.05 | 72.69 | 79.61 | 86.85 | 80.97 | 69.10 | 72.74 | 72.05 | **73.53** |

*Table 19.* Accuracy comparisons of different TTA methods on CIFAR-10-C dataset at damage level of 5, based on ResNet-50 backbone. The best results are highlighted in **boldface**, and the second ones are underlined.

| Method | CIFAR-10-C | | | | | | | | | | | | | | | Avg |
|---|---|---|---|---|---|---|---|---|---|---|---|---|---|---|---|---|
| | Gau. | Sho. | Imp. | Def. | Gla. | Mot. | Zoo. | Sno. | Fro. | Fog | Bri. | Con. | Ela. | Pix. | Jpe. | |
| Source (He et al., 2016) | 37.25 | 44.31 | 39.94 | 83.16 | 70.31 | 83.54 | 85.80 | 87.15 | 85.06 | 79.19 | 92.75 | 29.73 | 84.73 | 84.68 | 84.58 | 71.48 |
| BN (Schneider et al., 2020) | 0.00 | 0.00 | 0.00 | 0.00 | 0.00 | 0.00 | 0.00 | 0.00 | 0.00 | 0.00 | 0.00 | 0.00 | 0.00 | 0.00 | 0.00 | 0.00 |
| T3A (Iwasawa & Matsuo, 2021) | 47.84 | 52.78 | 51.52 | 83.16 | 73.06 | 83.35 | 85.66 | 87.04 | 84.96 | 79.53 | 92.72 | 36.44 | 84.49 | 85.72 | 84.81 | 74.21 |
| AdaNPC (Zhang et al., 2023) | 48.70 | 54.14 | 50.95 | 83.31 | 74.10 | 83.46 | 85.86 | 87.20 | 85.31 | 80.09 | 92.70 | 37.92 | 84.70 | 85.84 | 84.70 | 74.60 |
| TENT (Wang et al., 2020) | 37.25 | 44.31 | 39.94 | 83.16 | 70.31 | 83.54 | 85.80 | 87.15 | 85.05 | 79.19 | 92.75 | 29.73 | 84.73 | 84.68 | 84.58 | 71.48 |
| PLC (Lee, 2013) | 37.18 | 44.27 | 39.84 | 83.15 | 70.31 | 83.54 | 85.77 | 87.15 | 85.07 | 79.19 | 92.75 | 29.72 | 84.74 | 84.69 | 84.59 | 71.46 |
| EATA (Niu et al., 2022) | 46.55 | 48.34 | 31.91 | 86.30 | 69.31 | 84.78 | 86.56 | 88.62 | 87.25 | 80.32 | 93.05 | 45.84 | 84.87 | 86.99 | 84.29 | 73.67 |
| SAR (Niu et al., 2023) | 37.25 | 44.31 | 39.94 | 83.16 | 70.31 | 83.54 | 85.80 | 87.15 | 85.06 | 79.19 | 92.75 | 29.73 | 84.73 | 84.68 | 84.58 | 71.48 |
| TIPI (Nguyen et al., 2023) | 37.24 | 44.32 | 39.93 | 83.17 | 70.30 | 83.55 | 85.77 | 87.16 | 85.07 | 79.18 | 92.74 | 29.73 | 84.75 | 84.69 | 84.58 | 71.48 |
| TEA (Yuan et al., 2024) | 37.23 | 44.31 | 39.92 | 83.17 | 70.30 | 83.56 | 85.79 | 87.15 | 85.06 | 79.20 | 92.75 | 29.73 | 84.74 | 84.69 | 84.58 | 71.48 |
| TSD (Wang et al., 2023) | 37.17 | 44.22 | 39.80 | 83.18 | 70.35 | 83.58 | 85.80 | 87.16 | 85.08 | 79.20 | 92.75 | 29.70 | 84.74 | 84.70 | 84.59 | 71.47 |
| LinearTCA | 56.10 | 60.11 | **55.13** | 85.21 | **76.10** | 84.90 | 87.50 | 87.89 | 87.00 | 82.26 | 92.86 | 45.61 | 85.64 | 87.20 | **85.37** | 77.26 |
| LinearTCA + | **64.74** | **64.97** | 54.15 | **87.24** | 75.39 | **85.88** | **88.35** | **88.94** | **88.24** | **83.10** | **93.09** | **60.32** | **85.72** | **88.16** | 84.96 | **79.55** |

*Table 20.* Accuracy comparisons of different TTA methods on CIFAR-10-C dataset at damage level of 5, based on ViT-B/16 backbone. The best results are highlighted in **boldface**, and the second ones are underlined.

| Method | CIFAR-100-C | | | | | | | | | | | | | | | Avg |
|---|---|---|---|---|---|---|---|---|---|---|---|---|---|---|---|---|
| | Gau. | Sho. | Imp. | Def. | Gla. | Mot. | Zoo. | Sno. | Fro. | Fog | Bri. | Con. | Ela. | Pix. | Jpe. | |
| Source (He et al., 2016) | 10.46 | 12.49 | 3.36 | 34.44 | 23.63 | 38.10 | 42.67 | 39.25 | 33.01 | 32.84 | 55.78 | 11.55 | 46.48 | 34.88 | 46.15 | 31.01 |
| BN (Schneider et al., 2020) | 39.78 | 39.81 | 29.95 | 56.18 | 40.92 | 54.71 | 58.68 | 48.52 | 49.59 | 46.79 | 61.89 | 48.63 | 50.26 | 54.61 | 45.37 | 48.38 |
| T3A (Iwasawa & Matsuo, 2021) | 10.51 | 11.59 | 3.93 | 36.77 | 26.94 | 40.54 | 45.08 | 39.49 | 34.68 | 35.63 | 56.05 | 13.23 | 47.61 | 40.63 | 45.15 | 32.52 |
| AdaNPC (Zhang et al., 2023) | 10.01 | 10.69 | 3.64 | 33.57 | 24.38 | 37.00 | 41.06 | 35.88 | 31.45 | 32.28 | 52.19 | 12.29 | 43.32 | 37.21 | 40.50 | 29.70 |
| TENT (Wang et al., 2020) | 43.19 | 44.38 | 31.70 | 58.86 | 43.29 | 56.57 | 61.00 | 51.19 | 50.66 | 50.75 | 64.02 | 47.77 | 52.08 | 57.74 | 49.11 | 50.82 |
| PLC (Lee, 2013) | 39.65 | 39.47 | 30.25 | 56.31 | 40.70 | 54.50 | 58.88 | 48.56 | 49.37 | 46.73 | 62.04 | 48.68 | 50.44 | 54.20 | 45.45 | 48.35 |
| EATA (Niu et al., 2022) | 41.95 | 41.87 | 31.96 | 57.55 | 42.62 | 55.94 | 59.00 | 49.47 | 50.43 | 48.48 | 62.54 | 49.57 | 51.12 | 55.64 | 47.50 | 49.71 |
| SAR (Niu et al., 2023) | 44.07 | 45.12 | 33.37 | **59.80** | 43.69 | 57.21 | 61.15 | 51.70 | 51.97 | 51.49 | 63.90 | 50.46 | 52.64 | 57.97 | 49.52 | 51.60 |
| TIPI (Nguyen et al., 2023) | 44.04 | 45.11 | 32.86 | 57.89 | 43.85 | 55.87 | 60.08 | 52.16 | 51.69 | 49.38 | 63.40 | 44.24 | 51.43 | 57.42 | 49.76 | 50.61 |
| TEA (Yuan et al., 2024) | 43.78 | 43.43 | 32.68 | 58.20 | 42.62 | 56.30 | 60.67 | 50.84 | 51.32 | 50.16 | 63.87 | 49.95 | 51.78 | 56.60 | 47.83 | 50.67 |
| TSD (Wang et al., 2023) | 41.77 | 42.52 | 32.16 | 57.88 | 41.38 | 56.08 | 59.84 | 49.30 | 50.43 | 49.65 | 62.83 | 43.52 | 50.49 | 55.23 | 47.20 | 49.35 |
| LinearTCA | 13.98 | 16.45 | 5.42 | 38.96 | 29.15 | 42.56 | 46.30 | 42.40 | 39.41 | 39.56 | 56.78 | 15.33 | 49.51 | 42.56 | 47.07 | 35.03 |
| LinearTCA + | **44.70** | **45.77** | **33.76** | 59.77 | **44.45** | **57.41** | **61.49** | **52.25** | **52.52** | **51.92** | **64.25** | **51.18** | **53.28** | **58.68** | **49.81** | **52.08** |

*Table 21.* Accuracy comparisons of different TTA methods on CIFAR-100-C dataset at damage level of 5, based on ResNet-18 backbone. The best results are highlighted in **boldface**, and the second ones are underlined.

| Method | CIFAR-100-C | | | | | | | | | | | | | | | Avg |
|---|---|---|---|---|---|---|---|---|---|---|---|---|---|---|---|---|
| | Gau. | Sho. | Imp. | Def. | Gla. | Mot. | Zoo. | Sno. | Fro. | Fog | Bri. | Con. | Ela. | Pix. | Jpe. | |
| Source (He et al., 2016) | 17.23 | 19.42 | 9.77 | 35.34 | 31.87 | 39.15 | 41.98 | 41.99 | 38.68 | 32.00 | 54.56 | 11.18 | 47.57 | 42.51 | 47.02 | 34.02 |
| BN (Schneider et al., 2020) | 42.09 | 42.22 | 31.37 | 56.23 | 42.36 | 54.61 | 57.22 | 48.43 | 49.61 | 45.29 | 60.06 | 45.07 | 50.52 | 55.09 | 45.96 | 48.41 |
| T3A (Iwasawa & Matsuo, 2021) | 18.46 | 20.58 | 10.98 | 37.34 | 34.81 | 40.71 | 44.04 | 40.74 | 39.09 | 33.48 | 53.89 | 10.94 | 47.23 | 45.59 | 46.20 | 34.94 |
| AdaNPC (Zhang et al., 2023) | 17.41 | 18.85 | 10.53 | 35.26 | 31.43 | 37.49 | 40.99 | 37.25 | 35.78 | 30.65 | 49.75 | 10.68 | 43.74 | 41.78 | 42.53 | 32.27 |
| TENT (Wang et al., 2020) | 43.96 | 44.24 | 31.76 | **58.87** | 43.16 | **56.70** | **59.49** | 50.64 | 50.86 | 49.07 | 60.81 | 43.55 | **52.37** | 57.94 | 48.39 | 50.12 |
| PLC (Lee, 2013) | 41.80 | 42.50 | 31.57 | 55.98 | 42.62 | 54.60 | 57.27 | 48.35 | 48.83 | 45.28 | 60.03 | 44.77 | 50.47 | 55.23 | 46.34 | 48.38 |
| EATA (Niu et al., 2022) | 44.69 | 44.76 | 34.96 | 57.10 | 43.49 | 56.26 | 58.80 | 49.86 | 50.29 | 47.29 | 61.00 | 45.32 | 51.65 | 56.05 | 46.81 | 49.89 |
| SAR (Niu et al., 2023) | 44.59 | 44.64 | 34.57 | 58.26 | 43.55 | 56.41 | 58.62 | 50.08 | 50.74 | 47.77 | 61.39 | **46.76** | 51.49 | 56.85 | 48.07 | 50.25 |
| TIPI (Nguyen et al., 2023) | 46.12 | 46.31 | 34.13 | 57.48 | 43.46 | 55.63 | 58.51 | 51.32 | **52.45** | 48.56 | 61.05 | 40.80 | 51.28 | 57.93 | 49.48 | 50.30 |
| TEA (Yuan et al., 2024) | 44.64 | 45.79 | 34.71 | 57.63 | 43.66 | 56.11 | 58.37 | 50.18 | 50.21 | 48.86 | 61.11 | 45.59 | 51.21 | 56.46 | 48.61 | 50.21 |
| TSD (Wang et al., 2023) | 45.37 | 46.18 | 34.51 | 57.85 | 42.44 | 55.98 | 58.50 | 50.33 | 50.54 | **49.66** | 60.61 | 36.94 | 50.92 | 56.05 | 48.19 | 49.60 |
| LinearTCA | 21.90 | 24.46 | 12.80 | 39.80 | 36.53 | 42.66 | 45.80 | 43.03 | 42.66 | 36.47 | 55.13 | 12.97 | 49.49 | 47.41 | 48.09 | 37.28 |
| LinearTCA $^+$ | **47.29** | **48.95** | **36.13** | 57.60 | **44.46** | 55.68 | 58.80 | **53.31** | 52.11 | 48.68 | **61.78** | 41.87 | 51.49 | **58.48** | **50.99** | **51.17** |

*Table 22.* Accuracy comparisons of different TTA methods on CIFAR-100-C dataset at damage level of 5, based on ResNet-50 backbone. The best results are highlighted in **boldface**, and the second ones are underlined.

| Method | CIFAR-100-C | | | | | | | | | | | | | | | Avg |
|---|---|---|---|---|---|---|---|---|---|---|---|---|---|---|---|---|
| | Gau. | Sho. | Imp. | Def. | Gla. | Mot. | Zoo. | Sno. | Fro. | Fog | Bri. | Con. | Ela. | Pix. | Jpe. | |
| Source (He et al., 2016) | 21.71 | 24.74 | 19.53 | 62.41 | 43.14 | 61.13 | 67.65 | 66.34 | 67.48 | 54.03 | 77.43 | 33.26 | 60.09 | 60.48 | 56.17 | 51.71 |
| BN (Schneider et al., 2020) | 0.00 | 0.00 | 0.00 | 0.00 | 0.00 | 0.00 | 0.00 | 0.00 | 0.00 | 0.00 | 0.00 | 0.00 | 0.00 | 0.00 | 0.00 | 0.00 |
| T3A (Iwasawa & Matsuo, 2021) | 24.32 | 27.25 | 23.66 | 65.04 | 47.67 | 63.22 | 69.56 | 67.46 | 69.16 | 57.09 | 78.16 | 36.55 | 62.36 | 63.61 | 58.53 | 54.24 |
| AdaNPC (Zhang et al., 2023) | 22.75 | 25.77 | 22.65 | 63.54 | 46.46 | 62.02 | 68.99 | 66.78 | 68.50 | 56.45 | 77.60 | 35.34 | 61.63 | 62.87 | 56.82 | 53.21 |
| TENT (Wang et al., 2020) | 10.95 | 13.94 | 4.40 | 66.79 | 45.92 | 67.13 | 71.28 | 67.83 | 69.92 | 59.26 | 78.42 | 49.29 | 62.18 | 66.26 | 57.29 | 52.72 |
| PLC (Lee, 2013) | 21.66 | 24.71 | 19.49 | 62.42 | 43.16 | 61.15 | 67.61 | 66.34 | 67.53 | 54.04 | 77.44 | 33.29 | 60.10 | 60.49 | 56.17 | 51.71 |
| EATA (Niu et al., 2022) | 50.06 | 52.96 | 44.88 | 70.07 | 54.45 | 69.01 | 70.21 | 66.45 | 70.10 | 62.13 | 78.08 | 60.10 | 62.59 | 66.26 | 58.61 | 62.40 |
| SAR (Niu et al., 2023) | 16.59 | 18.07 | 9.89 | 67.86 | 47.37 | 67.31 | 71.48 | 67.99 | 70.19 | 60.58 | 78.17 | 52.90 | 61.29 | 66.11 | 58.56 | 54.29 |
| TIPI (Nguyen et al., 2023) | 7.95 | 9.85 | 3.77 | 67.08 | 45.89 | 66.96 | **71.98** | 68.01 | 70.63 | 59.47 | 78.24 | 47.70 | 62.37 | 67.37 | 58.17 | 52.36 |
| TEA (Yuan et al., 2024) | 10.99 | 17.39 | 8.09 | 66.54 | 45.55 | 65.24 | 70.78 | 67.06 | 69.09 | 58.30 | 76.44 | 45.15 | 61.60 | 64.82 | 57.56 | 52.31 |
| TSD (Wang et al., 2023) | 21.53 | 24.49 | 19.03 | 62.61 | 43.25 | 61.34 | 67.72 | 66.34 | 67.67 | 54.15 | 77.46 | 33.36 | 60.10 | 60.73 | 56.26 | 51.74 |
| LinearTCA | 27.46 | 30.02 | 25.33 | 65.29 | 47.98 | 64.26 | 69.91 | **68.32** | 70.01 | 58.49 | 78.16 | 39.42 | 62.74 | 65.09 | 58.82 | 55.42 |
| LinearTCA $^+$ | **51.98** | **54.92** | **46.74** | **71.00** | **56.07** | **69.73** | 71.06 | 67.56 | **71.01** | **63.93** | **78.61** | **62.35** | **63.42** | **67.73** | **59.49** | **63.71** |

*Table 23.* Accuracy comparisons of different TTA methods on CIFAR-100-C dataset at damage level of 5, based on ViT-B/16 backbone. The best results are highlighted in **boldface**, and the second ones are underlined.

| Method | ImageNet-C | | | | | | | | | | | | | | | Avg |
|---|---|---|---|---|---|---|---|---|---|---|---|---|---|---|---|---|
| | Gau. | Sho. | Imp. | Def. | Gla. | Mot. | Zoo. | Sno. | Fro. | Fog | Bri. | Con. | Ela. | Pix. | Jpe. | |
| Source (He et al., 2016) | 1.54 | 2.27 | 1.48 | 11.44 | 8.68 | 11.12 | 17.62 | 10.64 | 16.21 | 14.02 | 51.52 | 3.44 | 16.49 | 23.35 | 30.67 | 14.70 |
| BN (Schneider et al., 2020) | 13.65 | 14.84 | 14.17 | 11.95 | 13.04 | 23.34 | 33.89 | 29.18 | 28.42 | 40.80 | 58.11 | 12.09 | 38.92 | 44.35 | 37.08 | 27.59 |
| T3A (Iwasawa & Matsuo, 2021) | 1.61 | 2.35 | 1.65 | 10.57 | 8.20 | 10.12 | 17.38 | 11.03 | 16.14 | 15.19 | 49.23 | 3.24 | 18.00 | 23.37 | 30.31 | 14.56 |
| AdaNPC (Zhang et al., 2023) | 1.42 | 2.01 | 1.42 | 8.23 | 6.49 | 7.64 | 13.82 | 8.50 | 12.08 | 11.97 | 42.81 | 2.77 | 15.41 | 19.92 | 24.49 | 11.93 |
| TENT (Wang et al., 2020) | 23.45 | 25.71 | 24.08 | 18.79 | 20.90 | 33.54 | 42.85 | 39.64 | 32.95 | 50.36 | 60.13 | 10.68 | 48.81 | 51.96 | 46.98 | 35.39 |
| PLC (Lee, 2013) | 13.64 | 14.85 | 14.16 | 11.96 | 13.02 | 23.36 | 33.91 | 29.18 | 28.43 | 40.78 | 58.12 | 12.08 | 38.91 | 44.35 | 37.08 | 27.59 |
| EATA (Niu et al., 2022) | 28.24 | 30.16 | **28.88** | 25.30 | 25.74 | **36.61** | 43.71 | 41.80 | 36.42 | 50.87 | 59.12 | 31.75 | 49.10 | 52.33 | 47.82 | 39.19 |
| SAR (Niu et al., 2023) | 28.04 | 29.59 | 27.88 | 23.66 | 23.90 | 36.16 | 43.40 | 40.94 | **36.71** | **51.01** | **60.18** | 27.38 | 48.95 | **52.47** | **47.98** | 38.55 |
| TIPI (Nguyen et al., 2023) | 24.45 | 26.52 | 24.75 | 20.37 | 22.25 | 33.65 | 42.46 | 39.31 | 33.47 | 49.93 | 59.44 | 12.53 | 48.41 | 51.51 | 46.92 | 35.73 |
| TEA (Yuan et al., 2024) | 18.82 | 20.50 | 19.00 | 16.27 | 17.68 | 28.51 | 39.17 | 35.19 | 32.26 | 46.92 | 59.16 | 15.42 | 44.39 | 48.81 | 43.64 | 32.38 |
| TSD (Wang et al., 2023) | 15.60 | 16.99 | 16.13 | 15.59 | 15.41 | 28.69 | 38.07 | 32.92 | 30.01 | 45.90 | 58.69 | 7.62 | 41.06 | 47.47 | 41.52 | 30.11 |
| LinearTCA | 2.22 | 3.05 | 2.15 | 11.44 | 9.11 | 11.56 | 19.46 | 13.19 | 18.71 | 17.07 | 52.18 | 3.70 | 19.56 | 25.43 | 32.30 | 16.07 |
| LinearTCA $^+$ | **28.25** | **30.20** | 28.80 | **25.34** | **25.74** | 36.50 | **43.73** | **41.82** | 36.52 | 50.91 | 59.14 | **31.79** | **49.17** | 52.37 | 47.88 | **39.21** |

*Table 24.* Accuracy comparisons of different TTA methods on ImageNet-C dataset at damage level of 5, based on ResNet-18 backbone. The best results are highlighted in **boldface**, and the second ones are underlined.

| Method | ImageNet-C | | | | | | | | | | | | | | | Avg |
|---|---|---|---|---|---|---|---|---|---|---|---|---|---|---|---|---|
| | Gau. | Sho. | Imp. | Def. | Gla. | Mot. | Zoo. | Sno. | Fro. | Fog | Bri. | Con. | Ela. | Pix. | Jpe. | |
| Source (He et al., 2016) | 3.00 | 3.70 | 2.64 | 17.91 | 9.74 | 14.71 | 22.45 | 16.60 | 23.06 | 24.01 | 59.12 | 5.38 | 16.51 | 20.87 | 32.63 | 18.15 |
| BN (Schneider et al., 2020) | 16.32 | 17.09 | 16.97 | 15.23 | 15.54 | 26.64 | 39.38 | 34.46 | 33.45 | 48.43 | 65.67 | 17.08 | 44.62 | 49.49 | 40.47 | 32.06 |
| T3A (Iwasawa & Matsuo, 2021) | 2.97 | 3.38 | 2.65 | 17.05 | 9.37 | 13.69 | 22.63 | 16.98 | 22.83 | 25.34 | 57.68 | 5.05 | 18.41 | 20.31 | 32.36 | 18.05 |
| AdaNPC (Zhang et al., 2023) | 2.61 | 3.00 | 2.36 | 14.19 | 7.92 | 11.14 | 18.88 | 14.22 | 18.65 | 22.08 | 52.65 | 4.21 | 16.57 | 17.73 | 28.04 | 15.62 |
| TENT (Wang et al., 2020) | 26.70 | 28.90 | 28.01 | 25.01 | 24.72 | 38.66 | 48.84 | 46.04 | 40.49 | 57.04 | 67.96 | 24.42 | 53.73 | 57.72 | 51.54 | 41.32 |
| PLC (Lee, 2013) | 16.32 | 17.09 | 16.96 | 15.24 | 15.54 | 26.64 | 39.39 | 34.44 | 33.45 | 48.44 | 65.66 | 17.07 | 44.61 | 49.50 | 40.47 | 32.06 |
| EATA (Niu et al., 2022) | 35.38 | 37.75 | **36.34** | 33.29 | 32.74 | 47.51 | 53.07 | **52.53** | 46.30 | 60.54 | 68.06 | 43.82 | 58.42 | **61.15** | **55.72** | 48.17 |
| SAR (Niu et al., 2023) | 34.39 | 35.42 | 35.77 | 32.27 | 31.15 | 45.34 | 51.58 | 50.03 | 44.20 | 59.27 | **68.14** | 35.58 | 56.82 | 60.06 | 54.52 | 46.30 |
| TIPI (Nguyen et al., 2023) | 27.96 | 31.36 | 31.37 | 24.74 | 24.39 | 42.68 | 49.93 | 48.01 | 37.40 | 57.94 | 66.51 | 16.66 | 56.19 | 58.94 | 53.90 | 41.87 |
| TEA (Yuan et al., 2024) | 22.51 | 23.94 | 22.70 | 20.70 | 21.00 | 36.06 | 46.93 | 44.91 | 39.37 | 56.06 | 67.01 | 22.79 | 52.65 | 56.86 | 49.99 | 38.90 |
| TSD (Wang et al., 2023) | 18.90 | 18.80 | 19.63 | 18.31 | 17.49 | 31.12 | 44.27 | 39.74 | 36.02 | 53.38 | 67.00 | 13.06 | 47.97 | 54.54 | 45.95 | 35.08 |
| LinearTCA | 3.35 | 4.39 | 3.08 | 17.67 | 10.01 | 15.12 | 22.89 | 19.21 | 25.61 | 27.30 | 59.50 | 5.81 | 20.39 | 21.72 | 34.11 | 19.34 |
| LinearTCA $^+$ | **35.44** | **37.78** | 36.29 | **33.37** | **32.80** | **47.63** | **53.32** | 52.45 | **46.32** | **60.55** | 68.04 | **43.90** | **58.61** | 61.08 | 55.71 | **48.22** |

*Table 25.* Accuracy comparisons of different TTA methods on ImageNet-C dataset at damage level of 5, based on ResNet-50 backbone. The best results are highlighted in **boldface**, and the second ones are underlined.

| Method | ImageNet-C | | | | | | | | | | | | | | | Avg |
|---|---|---|---|---|---|---|---|---|---|---|---|---|---|---|---|---|
| | Gau. | Sho. | Imp. | Def. | Gla. | Mot. | Zoo. | Sno. | Fro. | Fog | Bri. | Con. | Ela. | Pix. | Jpe. | |
| Source (He et al., 2016) | 35.09 | 32.16 | 35.88 | 31.42 | 25.31 | 39.45 | 31.55 | 24.47 | 30.13 | 54.74 | 64.48 | 48.98 | 34.20 | 53.17 | 56.45 | 39.83 |
| BN (Schneider et al., 2020) | 0.00 | 0.00 | 0.00 | 0.00 | 0.00 | 0.00 | 0.00 | 0.00 | 0.00 | 0.00 | 0.00 | 0.00 | 0.00 | 0.00 | 0.00 | 0.00 |
| T3A (Iwasawa & Matsuo, 2021) | 27.87 | 28.15 | 30.27 | 32.42 | 27.00 | 40.66 | 33.52 | 25.95 | 30.76 | 56.32 | 64.85 | 50.20 | 37.99 | 53.75 | 57.01 | 39.78 |
| AdaNPC (Zhang et al., 2023) | 30.01 | 26.86 | 30.98 | 28.19 | 23.40 | 36.38 | 29.65 | 21.18 | 26.59 | 52.78 | 61.24 | 44.53 | 34.44 | 50.50 | 54.98 | 36.78 |
| TENT (Wang et al., 2020) | 51.19 | 50.00 | 52.48 | 47.35 | 42.95 | 54.40 | 45.19 | 7.43 | 16.27 | 64.87 | 70.90 | 64.35 | 25.97 | 63.35 | 63.48 | 48.01 |
| PLC (Lee, 2013) | 29.10 | 29.61 | 31.58 | 31.19 | 25.10 | 39.33 | 31.45 | 24.58 | 30.18 | 54.33 | 64.47 | 48.44 | 34.17 | 52.52 | 55.11 | 38.74 |
| EATA (Niu et al., 2022) | 56.63 | 56.20 | 57.49 | 56.13 | 57.13 | 62.26 | 62.89 | 64.02 | 62.77 | 73.71 | 77.07 | 70.34 | 67.39 | 71.42 | 69.96 | 64.36 |
| SAR (Niu et al., 2023) | 54.90 | 55.82 | 56.68 | 55.94 | 55.61 | 62.47 | 58.11 | 17.20 | 34.16 | 71.85 | **77.14** | 63.48 | 65.75 | 71.45 | 68.52 | 57.94 |
| TIPI (Nguyen et al., 2023) | 56.76 | **56.70** | **58.45** | 55.90 | 56.10 | 61.75 | 19.67 | 1.88 | 4.51 | 64.18 | 75.91 | 69.24 | 6.37 | 70.05 | 69.97 | 48.50 |
| TEA (Yuan et al., 2024) | 39.46 | 38.72 | 41.90 | 24.47 | 28.03 | 42.01 | 33.46 | 13.43 | 33.46 | 53.89 | 66.09 | 58.66 | 34.95 | 55.65 | 56.41 | 41.37 |
| TSD (Wang et al., 2023) | 36.78 | 33.68 | 37.61 | 32.28 | 26.36 | 40.93 | 32.65 | 25.20 | 31.49 | 56.17 | 65.65 | 54.27 | 35.10 | 54.61 | 57.17 | 41.33 |
| LinearTCA | 30.45 | 30.77 | 32.83 | 33.50 | 27.47 | 42.14 | 34.91 | 26.98 | 32.85 | 57.65 | 64.88 | 56.65 | 38.18 | 54.08 | 57.22 | 41.37 |
| LinearTCA $^+$ | **56.92** | 56.47 | 57.63 | **56.52** | **57.56** | **62.65** | **63.62** | **64.53** | **63.30** | **74.06** | 77.11 | **70.64** | **67.82** | **71.65** | **70.19** | **64.71** |

*Table 26.* Accuracy comparisons of different TTA methods on ImageNet-C dataset at damage level of 5, based on ViT-B/16 backbone. The best results are highlighted in **boldface**, and the second ones are underlined.

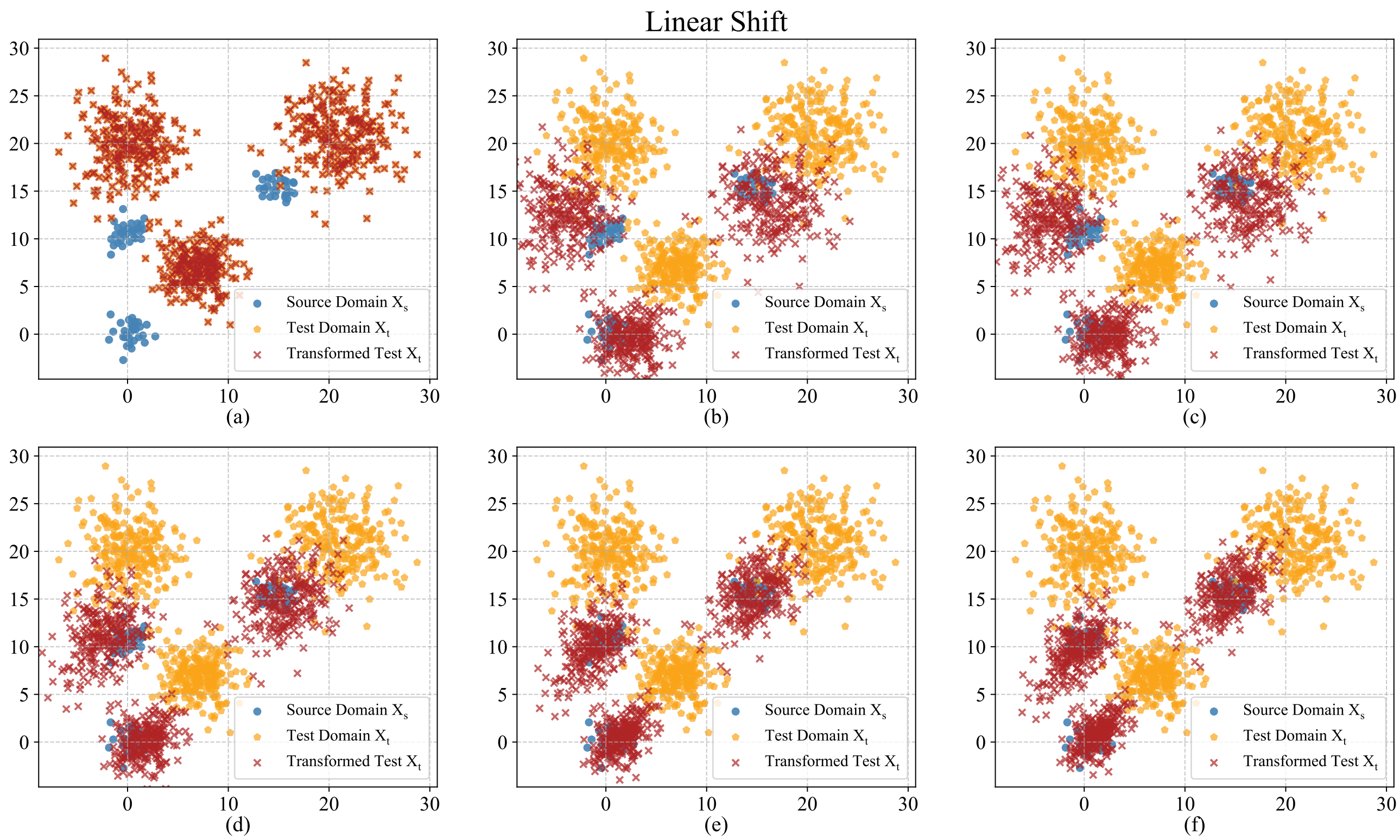

*Figure 5.* Adaptation process of LinearTCA to datasets with linear shifts.

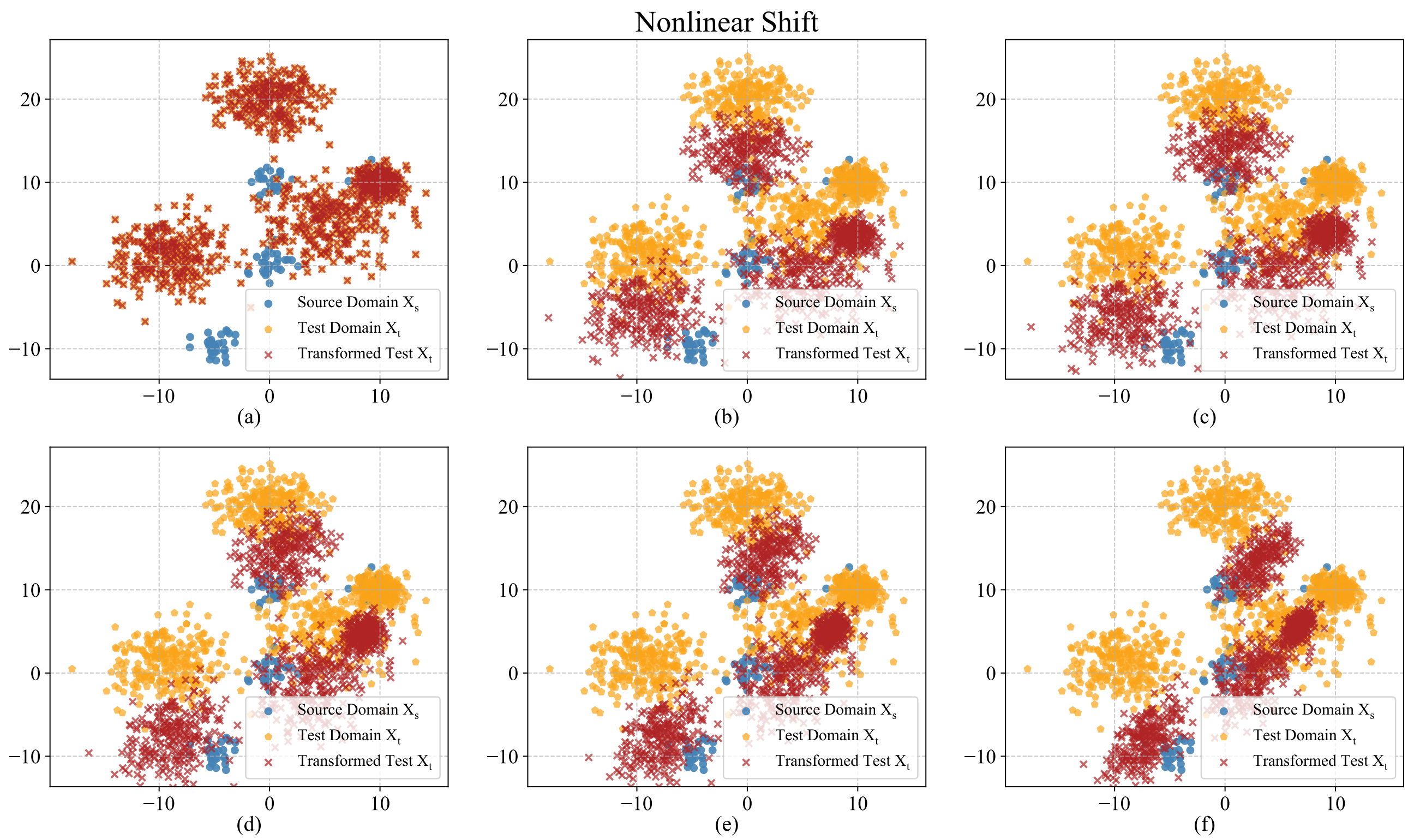

*Figure 6.* Adaptation process of LinearTCA to datasets with nonlinear shifts