# OpenReview forum: "Test-time Correlation Alignment"
_ICML.cc/2025/Conference — ICML 2025 poster_

### Official Review · Reviewer_Knrt · 2025-03-08

**Overall Recommendation:** 4

**Summary:**

This paper proposes Test-time Correlation Alignment (TCA), a novel method to address the challenges of test-time adaptation (TTA) in deep neural networks. TCA aims to enhance model performance on out-of-distribution test data by aligning feature correlations between high-certainty test instances and the test domain, without requiring access to source data. The authors introduce two algorithms, LinearTCA and LinearTCA+, which achieve correlation alignment through linear transformations without model updates. LinearTCA is efficient and memory-friendly, while LinearTCA+ serves as a plug-and-play module to boost existing TTA methods. Experiments across various benchmarks and backbones demonstrate that TCA significantly improves accuracy, efficiency, and resistance to domain forgetting compared to state-of-the-art TTA methods. Theoretical analysis and ablation studies further validate the effectiveness of TCA in reducing test classification error and adapting to distribution shifts.

**Claims And Evidence:**

YES

**Essential References Not Discussed:**

No

**Experimental Designs Or Analyses:**

* I recommend adding baselines such as PLC (using pseudo-labels to update the linear classifier) [1], T3A [2], and AdaNPC [3], which are also computation-light methods.

* The result of TSD [4] with ViT-B/16 on PACS is very low, which is very different from the original paper.

* There is a lack of large-scale datasets, such as DomainNet [5] and ImageNet-C [6].

* Batch size has an effect inTTA . I am curious about the performance under a small batch size because small batch sizes. It is better to show performance and estimation error between pseudo-source and source under difference batch size



**Reference**
[1]  Pseudo-label: The simple and efficient semi-supervised learning method for deep neural networks. 2013
[2] Test-time classifier adjustment module for model-agnostic domain generalization. NeurIPS 2021
[3] AdaNPC: Exploring Non-Parametric Classifier for Test-Time Adaptation. ICML 2023
[4] Feature Alignment and Uniformity for Test Time Adaptation. CVPR 2023
[5] Moment Matching for Multi-Source Domain Adaptation. ICCV 2019
[6] Benchmarking Neural Network Robustness to Common Corruptions and Perturbations. ICLR 2019

**Methods And Evaluation Criteria:**

* Do authors try to use more layers for transformation $W$, e.g. two linear layers with non-linear activation ? Does it will better performance?

See other sections for more comments.

**Other Comments Or Suggestions:**

Minor typo:
* inconsisent expression, e.g. Eq.12 in Line 189. It should be Eq.(12) which is consistent with others.

**Other Strengths And Weaknesses:**

See other sections

**Questions For Authors:**

See other sections

**Relation To Broader Scientific Literature:**

Previous works mainly study instance-wise methods in TTA. This paper presents another perspective: correlation alignment. It aligns the correlation of the target domain and the source domain. Due to the lack of source domain data, the paper proposes using a pseudo-source domain to estimate the correlation of the source domain.

**Theoretical Claims:**

Yes. I have carefully checked the correctness of proofs.

---

> ### Author Rebuttal · Authors · 2025-04-01
>
> We thank Reviewer Knrt for the valuable comments. (**Click [https://anonymous.4open.science/r/2025ICML_TCA_Rebuttal/README.md](https://anonymous.4open.science/r/2025ICML_TCA_Rebuttal/README.md) for Rebuttal Figures and Tables.**)
>
> ---
>
> ## Q1. Try to use more layers for transformation.
>
> Following the suggestion, we extend LinearTCA/LinearTCA+ by introducing MLP-based transformations with two (MLP2) and three (MLP3) layers. As shown in **Rebuttal Table 4**, incorporating nonlinear activations in deeper architectures leads to further performance improvements. As noted in the Conclusion of the original paper, future work may explore nonlinear transformations to enhance correlation alignment further. We welcome any suggestions from the reviewers regarding additional methods for TCA.
>
> ---
>
> ## Q2 & Q4: Comparisons with More Methods and Large-Scale Datasets
>
> As suggested, we add comparisons with PLC [1] and T3A [2] on DomainNet [4] and ImageNet-C [5]. The results in **Rebuttal Tables 5-8** demonstrate that:
>
> - LinearTCA+ consistently improves accuracy across different backbones and on both large-scale datasets.
> - LinearTCA performs well on DomainNet, outperforming most methods. However, on ImageNet-C, while LinearTCA improves upon the source model, it lags behind other methods. This is primarily because LinearTCA employs only linear transformations, whereas ImageNet-C contains 15 different types of corruptions, making purely linear adjustments insufficient.
> - PLC and T3A exhibit good efficiency and memory usage. LinearTCA remains the most efficient among all methods.
>
> It is noted that AdaNPC [3] modifies the loss function during training (using kNN loss) and uses source domain data at test time, which does not strictly follow the TTA setting. Given the difficulty of modifying the original code within a short timeframe, we plan to incorporate AdaNPC in the final version after the necessary code modifications.
>
> ---
>
> ## Q3: TSD [6] with ViT-B/16 on PACS Shows Large Degradation
>
> We observed a significant accuracy drop for TSD on PACS (*S domain: 78.77 → 58.63*, see Appendix Table 7) and conducted an in-depth analysis to investigate this issue. Our findings are:
>
> - We confirmed again that the hyperparameters provided in Appendix Table 7 correctly reproduce the reported results.
> - The reason is that **TSD is highly sensitive to hyperparameter tuning on the PACS dataset**, and our tuning strategy differs from TSD’s:
>    (1) Our approach: Tune hyperparameters on a single domain and apply them directly to other domains.
>    (2) TSD's original approach: Tune hyperparameters on the source domain before adaptation.
>
> - To address the reviewer’s concern, we conducted extensive hyperparameter tuning for TSD on the S domain (see **Rebuttal Table 9**). The results indicate that **our method consistently outperforms TSD regardless of TSD’s hyperparameters** (TCA 83.05 v.s. TSD 81.73).
>
> ---
>
> ## Q5. Performance and Estimation Error between Pseudo-Source and Source under Different Batch Sizes
>
> As presented in **Rebuttal Table 10**:
>
> 1. Even with a batch size of 1, LinearTCA still outperforms the source model by 2.16%.  Additionally, LinearTCA+ robustly improves the baseline method TEA across different batch sizes.
>    - This is primarily because the covariance matrix of the test domain is calculated incrementally. The estimated covariance matrix progressively converges to the true covarianc of the entire domain.
>    - When the batch size is small, the main effect is on the predictions made by the initial arriving samples. As more samples are processed, the impact of the batch size diminishes.  Therefore, in practice, if the total number of samples is large enough, the effect of the batch size becomes relatively insignificant.
>
> 2. The estimation error of the pseudo-source domain is independent of the batch size.
>    - This is because the pseudo-source domain is also calculated incrementally, and unlike the test domain, only a few dozen high-confidence samples (as shown in Figure 4 (c) of the original paper) are needed to achieve good performance.
>
> We appreciate this valuable suggestion and will make this point clearer in the final version.
>
> ---
>
> ## Q6. Minor Typo
>
> We apologize for any confusion caused by our notation. We will carefully review the entire manuscript to ensure clarity and consistency in the notation.
>
> Thank you for your attention to detail.
>
> ---
>
> ## Reference
> [1] Pseudo-label: The simple and efficient semi-supervised learning method for deep neural networks. 2013
> [2] Test-time classifier adjustment module for model-agnostic domain generalization. NeurIPS 2021
> [3] AdaNPC: Exploring Non-Parametric Classifier for Test-Time Adaptation. ICML 2023
> [4] Moment Matching for Multi-Source Domain Adaptation. ICCV 2019
> [5] Benchmarking Neural Network Robustness to Common Corruptions and Perturbations. ICLR 2019
> [6] Feature Alignment and Uniformity for Test Time Adaptation. CVPR 2023

---

> > ### Comment · Reviewer_Knrt · 2025-04-02
> >
> > Thank you for your response. Most of my concerns have been addressed, so I’ll increase my score to a 4.

---

> > > ### Author Response · Authors · 2025-04-03
> > >
> > > Dear Reviewer Knrt:
> > >
> > > Thank you for your feedback. We are pleased to hear that our responses have addressed most of your concerns and the update on the score. Your attention to detail and constructive comments are truly invaluable to us. We highly appreciate it.
> > >
> > > Wishing you all the best,
> > > Authors

---

### Official Review · Reviewer_cY7r · 2025-03-10

**Overall Recommendation:** 3

**Summary:**

The paper investigates the feasibility of performing correlation alignment during test time. Specifically, the authors:
1. Identifies the limitation of current TTA methods, and proposes Correlation Alignment (CORAL) as a potential soluton.
2. Introduces a strategy to construct a pseudo-source domain by sampling high-confidence target data to address the challenge of absent source domain during testing.
3. Provides theoretical justification of this pseudo-source approach under Lipschitz assumption.
4. Conducts experiments to: (1) validate the theoretical findings, and (2) demonstrate superior performance compared to existing TTA methods, yielding promising results.

**Claims And Evidence:**

While the throretical and emperical claims in the paper appear sound, I have concerns regarding the introduction of CORAL in the context of TTA. Specifically, Section 2.3 seems to lack detailed discussion and verification. I'd like to hear more evidence demonstrating the necessity of incorporating CORAL into TTA tasks.

**Essential References Not Discussed:**

N/A

**Experimental Designs Or Analyses:**

The experiment design appears adequate. However, the models and benchmark considered, such as ResNet18 and ViT-B/16,might be somewhat outdated, which limits its applicability in real scenario. Given that the scenario of "lack of access to source domain data" aligns closely with the constraints of closed source foundation models that people often use today (e.g. CLIP), would the authors be interested in evaluate the proposed method on such a benchmark?

**Methods And Evaluation Criteria:**

The evaluation dataset and benchmark for domain adaptation appears robust.

**Other Comments Or Suggestions:**

1. Typo, the 7th line of section 2.2: n_s should be n_t.
2. The authors should standardize the usage of $\hat{\Sigma_s}$ and $\Sigma_s$. Sometimes they represents pseudo-source interchangeably, which is confusing.

**Other Strengths And Weaknesses:**

N/A

**Questions For Authors:**

1. For experiment part, I'm curious about the upper performance bound for TCA under two scenarios: (a) directly fine-tuning on target distribution, and (b) applying LinearTCA and LinearTCA+ sampled from real source distribution.
2. As indicated in "Claims And Evidence" section, I'd like to hear discussion/evidence of the necessity of bringing CORAL.

**Relation To Broader Scientific Literature:**

The paper is well-presented with a clear narrative. Its approach to test-time distribution alignment could offer methodologies to complement existing test-time adaptation toolkits.

**Theoretical Claims:**

I checked the derivation of Theorem 3.5 and Theorem 3.6, their derivation to Corollary 3.7, and the consistency with experiments in section 5.2. Appears adequate.

---

> ### Author Rebuttal · Authors · 2025-04-01
>
> We sincerely appreciate Reviewer cY7r’s insightful comments and constructive feedback. Below, we address each concern in detail. (**Click [https://anonymous.4open.science/r/2025ICML_TCA_Rebuttal/README.md](https://anonymous.4open.science/r/2025ICML_TCA_Rebuttal/README.md) for Rebuttal Figures and Tables.**)
>
> ---
>
> ## Q1. Evaluation on closed-source foundation models (e.g., CLIP)
>
> As suggested, we conduct additional experiments incorporating CLIP on PACS, OfficeHome, and VLCS. The experimental setup and implementation details follow [1]. As shown in **Rebuttal Table 2**, our method consistently outperforms all baselines. Specifically:
>
> - Compared to other methods, both LinearTCA+ and LinearTCA achieve superior performance, with improvements of 1.28%, 2.08%, and 2.85% on the three datasets, respectively. This is because CLIP computes image-text similarity directly, whereas our TCA methods explicitly align the embedding distributions with the source domain, making them particularly effective in multi-modal models like CLIP.
> - The performance of LinearTCA+ and LinearTCA is similar, though LinearTCA+ achieves slightly better results. This suggests that even a simple correlation alignment method can significantly improve performance on widely used models like CLIP. Moreover, it demonstrates that LinearTCA+ functions as a plug-and-play module that enhances various adaptation methods when used as a complement.
>
> We appreciate this valuable suggestion and will incorporate these results and discussions in the final version for clarity.
>
> ---
>
> ## Q2. Upper performance bound for TCA
>
> To evaluate the upper bound of our method, we conducted two additional experiments (see **Rebuttal Table 3**):
>
> (a) Fine-tuning directly on the target distribution.
> (b) Applying LinearTCA and LinearTCA+ with real source distributions.
>
> - Compared to the original LinearTCA+, approach (b) further enhances performance, achieving improvements of 0.38% on PACS and 1.03% on OfficeHome.
> - Similarly, both approaches (a) and (b) outperform the original LinearTCA across multiple domains. Notably, on the OfficeHome dataset, even the relatively simple LinearTCA (b) surpasses TCA(a), underscoring the crucial role of source domain information in domain adaptation. This further reinforces the necessity of approximating the source distribution in TCA.
>
> ---
>
> ## Q3. Why integrating CORAL into TTA
>
> We clarify that incorporating CORAL into test-time adaptation (TTA) is supported by **prior research, empirical observations, theoretical analysis, and experimental results**. Specifically:
>
> - **Extensive studies** (see Appendix A.1. of the original paper) have demonstrated that correlation alignment (CORAL) is both effective and efficient for domain adaptation (DA). Since TTA is a subfield of DA, it is a natural extension to apply CORAL within TTA frameworks.
> - **Empirical observations** (see Section 1, line 066, and Figure 1 of the original paper) indicate that correlation distance increases during domain shifts, further motivating the need for correlation alignment in TTA.
> - **Our theoretical analysis** (see Theorem 3.6 of the original paper) shows that the generalization error in TTA includes second-order terms, and CORAL can directly mitigates them by aligning second-order statistics.
> - **Experimental results** (see Section 5.2 of the original paper) validate our theoretical findings—pseudo-source domains constructed from high-confidence samples closely approximate the true source distribution, and applying CORAL in this context leads to both effective and efficient TTA.
> - **Furthermore**, since source domain data is unavailable in TTA scenarios, an important research problem is exploring how to effectively apply CORAL in this setting.
> ### Summary:
> Our motivation stems from **prior research in domain adaptation (DA)** and **our own empirical observations**, and it is rigorously validated through both **theoretical analysis** and **experimental results**. These discussions will be included in Section 2.3 of the final version.
>
> ---
>
> ## Q4: Typographical Error
>
> We apologize for any confusion in our notation. To clarify, $\hat{\Sigma}_s$ represents the estimated pseudo-source domain covariance, while $\Sigma_s$ denotes the true source domain covariance. We will carefully review the entire manuscript to ensure clarity and consistency in notation.
>
> ---
> We appreciate these valuable insights and will incorporate additional discussions and refinements in the final version of the paper.
> ## Reference
> [1] WATT: Weight Average Test-Time Adaptation of CLIP. NeurIPS 2024

---

> > ### Comment · Reviewer_cY7r · 2025-04-06
> >
> > Thanks the authors to address my concerns. I'll change my evaluation to weak accept.

---

> > > ### Author Response · Authors · 2025-04-06
> > >
> > > Dear Reviewer cY7r:
> > >
> > > Thank you for your feedback and evaluation updating. Your insightful comments have significantly contributed to improving our work. We sincerely appreciate the time and effort you dedicated to reviewing our work.
> > >
> > > Wishing you all the best,
> > > Authors

---

### Official Review · Reviewer_SZVF · 2025-03-16

**Overall Recommendation:** 3

**Summary:**

The paper introduces TCA, which addresses some limitations of existing test-time adaptation (TTA) methods, such as neglecting feature correlation alignment and relying on backpropagation. TCA aligns feature correlations between test data and pseudo-source domains without accessing the source data. Two methods are proposed: LinearTCA, which uses linear transformations for correlation alignment, and LinearTCA+, a module to extend other TTA methods. The approach is supported by theoretical analysis and evaluated on benchmarks such as OfficeHome, PACS and CIFAR-10/100-C, showing non-trivial performance gains with reduced computational overhead.

**Claims And Evidence:**

The below claims have been supported by the experiment results in the manuscript.
1. TCA improves TTA by aligning feature correlations rather than just instance-wise features.
2. High-certainty test instances can approximate the source domain's feature correlation.
3. LinearTCA achieves competitive or superior performance compared to SOTA TTA methods.
4. LinearTCA significantly reduces memory and computational overhead.

**Essential References Not Discussed:**

N/A

**Experimental Designs Or Analyses:**

The experimental designs in this manuscript are enough and resonable.

**Methods And Evaluation Criteria:**

The paper evaluates TCA on:
1. Datasets: PACS, OfficeHome, CIFAR-10C, CIFAR-100C.
2. Backbones: ResNet-18, ResNet-50, ViT-B/16.
3. Metrics: Classification accuracy, correlation distance, GPU memory consumption, computation time, and resistance to forgetting.

**Other Comments Or Suggestions:**

It's better for authors to address the issues raised in the weaknesses to further improve the quality of the manuscript.

**Other Strengths And Weaknesses:**

**Weaknesses:**

1. Although the proposed method sounds reasonable, the learning rate used in the proposed method is variable across different domains, which is sensitive for a test-time methodology since no prior knowledge could be known in advance.

2.  The authors mentioned that they used a small value to obtain the pseudo-source correlation, such as 10. However, to obtain a full-rank or more stable covariance matrix, a larger number of samples relative to the feature dimension is usually required. By using 10 samples, I can imagine that only 10 eigenvalues will survive the decomposition. This worries me as the fit could be biased towards a few confident samples. Is this because only a few samples are enough to approximate the correlation? or does using more samples generally degrade performance due to their corrupted features?

**Questions For Authors:**

See the weaknesses above.

**Relation To Broader Scientific Literature:**

This work builds on and extends prior research in Test-Time Adaptation, Memory-Efficient Test-Time Adaptation.

**Theoretical Claims:**

1. The key theoretical contributions establish that aligning pseudo-source correlation with test correlation reduces classification error bounds (Theorem 3.6, Corollary 3.7).
2. The proofs appear rigorous, with clear assumptions (Lipschitz continuity, strong density conditions) and mathematical derivations supporting TCA's effectiveness.

---

> ### Author Rebuttal · Authors · 2025-04-01
>
> We sincerely thank Reviewer SZVF for the constructive and valuable comments. The concerns are addressed as follows. (**Click [https://anonymous.4open.science/r/2025ICML_TCA_Rebuttal/README.md](https://anonymous.4open.science/r/2025ICML_TCA_Rebuttal/README.md) for Rebuttal Figures and Tables.**)
>
> ---
>
> ## Q1: Concerns about the learning rate (lr)
>
> We acknowledge the reviewer’s concerns and clarify the **misunderstandings** as follows:
>
> - **Fixed learning rate:** Our method maintains a fixed learning rate $ lr = 10^{-3} $ across all datasets and domains. The only adjustable parameter in our method is the number of pseudo-source samples $k$ (as stated in Section 5.4, line 439 of the original paper). Any observed lr variability applies only to certain baseline methods, not to ours.
> - **Hyperparameter tuning is dataset-specific, not domain-specific:** The hyperparameter tuning in our experimental setup is performed across datasets, not across domains. Specifically, for all TTA methods, we first tune hyperparameters on the default domain (e.g., domain A in PACS) and then apply them directly to other domains (e.g., P, C, and S in PACS). The details of this setup are provided in the Appendix D.3, line 1040 of the original paper.
> - **Consistency with prior work:** Tuning hyperparameters at the dataset level is a widely adopted experimental protocol in TTA research [1-8].  Specifically, Tent [1], EATA [2], SAR [3], TIPI [4], and TEA [5] select hyperparameters from the source domain during training for each dataset, whereas T3A [6], TSD [7], and AdaNPC [8] choose the optimal hyperparameters from the validation set of each training dataset.
> - **Fixed hyperparameter experiment:** We fully agree with the reviewer that, in real-world scenarios, prior knowledge is often unavailable. To address this concern, we conducted additional experiments using fixed hyperparameters for all methods across two datasets. As shown in **Rebuttal Table 1**, our TCA method demonstrates strong robustness and generalization capabilities, even in the absence of prior knowledge. In contrast, some methods experienced significant performance degradation when applied across different datasets.
>
> We will improve the clarity of Appendix D.3 in the final version to better communicate these details.
>
> ---
>
> ## Q2: Concerns about the number of pseudo-source samples
>
> - **Addressing the full-rank issue:** This concern precisely aligns with why we suggest using gradient descent to solve for the transformation matrix $W $ (as stated in Section 4.2, line 327). In our experiments, we set a fixed learning rate $ lr = 10^{-3} $ and use 20 optimization steps to obtain $ W $.
> - **Effect of pseudo-source sample quantity:** A small number of high-confidence samples are sufficient to approximate the correlation structure effectively. Increasing the number of low-confidence samples often degrades performance.
>
>   As illustrated in **Rebuttal Figure 1**, the correlation distance in the pseudo-source domain initially decreases but then increases as more samples are included. This occurs because, as additional samples are added, their confidence levels drop, leading to a growing discrepancy between the computed pseudo-source domain and the true source domain.
>
>   These findings align with our theoretical results (line 200 in Theorem 3.5 of the original paper), demonstrating that high-confidence samples are closer to the true source distribution.
>
> We appreciate these valuable insights and will incorporate additional discussions and refinements in the final version of the paper.
>
> ## Reference
>
> [1] Tent: Fully test-time adaptation by entropy minimization. ICLR 2021
> [2] Efficient test-time model adaptation without forgetting. ICML 2022
> [3] Towards stable test-time adaptation in dynamic wild world. ICML 2023
> [4] Tipi: Test time adaptation with transformation invariance. CVPR 2023
> [5] Tea: Test-time energy adaptation. CVPR 2024
> [6] Test-time classifier adjustment module for model-agnostic domain generalization. NeurIPS 2021
> [7] Feature alignment and uniformity for test time adaptation. CVPR 2023
> [8] AdaNPC: Exploring Non-Parametric Classifier for Test-Time Adaptation. ICML 2023

---

### Decision · Program_Chairs · 2025-05-01

**Decision:**

Accept (poster)

**Comment:**

This paper proposes Test-time Correlation Alignment (TCA), a theoretically grounded and efficient approach to Test-Time Adaptation (TTA) that aligns feature correlations between high-confidence test samples and the test domain without accessing source data. Reviewers found the method well-motivated and appreciated its empirical gains, low computational cost, and theoretical contributions, though initial concerns included hyperparameter sensitivity, limited evaluation on foundation models, and small pseudo-source sample size. The rebuttal convincingly addressed these points with new experiments on CLIP and large-scale datasets, detailed clarifications on experimental protocols, and ablations on sample count and batch size. All reviewers acknowledged the improvements and raised their scores to weak accept or accept. Overall, the paper is a solid contribution to TTA with promising practical value. For further improvement, the authors are encouraged to expand large-scale evaluations and clarify the robustness of TCA under varying test conditions.